# Topography and human pressure in mountain ranges alter expected species responses to climate change

Paul R. Elsen [1,2✉], William B. Monahan[3] & Adina M. Merenlender[1]

Climate change is leading to widespread elevational shifts thought to increase species extinction risk in mountains. We integrate digital elevation models with a metric of human pressure to examine changes in the amount of intact land area available for species undergoing elevational range shifts in all major mountain ranges globally ($n = 1010$). Nearly 60% of mountainous area is under intense human pressure, predominantly at low elevations and mountain bases. Consequently, upslope range shifts generally resulted in modeled species at lower elevations expanding into areas of lower human pressure and, due to complex topography, encountering more intact land area relative to their starting position. Such gains were often attenuated at high elevations as land-use constraints diminished and topographic constraints increased. Integrating patterns of topography and human pressure is essential for accurate species vulnerability assessments under climate change, as priorities for protecting, connecting, and restoring mountain landscapes may otherwise be misguided.

[1] Department of Environmental Science, Policy, and Management, University of California, Berkeley, Berkeley, CA 94720, USA. [2] Wildlife Conservation Society, 2300 Southern Boulevard, Bronx, NY 10460, USA. [3] USDA Forest Service, Forest Health Protection, Fort Collins, CO 80526, USA. ✉email: pelsen@wcs.org

Climate change is causing widespread elevational range shifts in plant and animal species in mountainous regions[1–6]. Such elevational range shifts are often thought to be associated with an increased risk of extinction as topographic constraints impose significant reductions in the amount of area available for species following range shifts[7,8]. However, owing to complex topography in mountain ranges, such topographic constraints can occur at virtually any position along elevational gradients[9]. For instance, topographic constraints are roughly uniform along elevations in mountain ranges with unimodal declines in surface area ('pyramid' mountains), such as the European Alps, whereas they are greatest at high elevations in mountain ranges with mid-elevation peaks of surface area ('diamond' mountains), such as the Rocky Mountains of North America[9]. Few studies account for topographic patterns using high-resolution data, which could lead to inaccurate expectations of where species may experience range contractions following climate change[10].

Another limitation in many approaches to forecasting changes in available area for species persistence under climate change involves accounting for how human pressures limit species distributions and movement required to shift their range to adapt to change[11]. Human pressure on montane landscapes is predominantly through resource extraction, infrastructure development, and habitat conversion. Indeed, habitat conversion for agriculture, pasture, and cropland is extensive in mountainous regions[12], is a leading driver of biodiversity loss globally[13], and is often expected to exacerbate the negative effects of climate change[14]. However, because human pressure is typically biased towards low elevations[15] and protection towards high elevations in mountain ranges globally[16], higher elevation lands could provide refuge from human activities[11,15].

Ultimately, accurate assessments of changes in habitable area for species undergoing upslope range shifts rely on integrating fine-scale topography with current land use patterns. Predictions of area changes due to upslope movement under climate change may be inaccurate when omitting high-resolution topography in assessments[17–19]. While mountainous regions are generally considered well-protected[16,20], there is significant variation in protection across continents and ecoregions[21], and even greater variation across individual mountain ranges[16], the spatial scale most aligned with typical montane species distributions[22]. Moreover, many of these regions have faced increasing pressure from human populations over recent decades[23], including within protected areas[24]. Consequently, population responses and threats to extinction arising from elevational range shifts will likely be highly context specific and potentially deviate from expectations derived from considerations of topographic constraints or habitat availability in isolation.

Despite the recognized importance of topography and land-use in constraining species distributions[11], to date there has been no global assessment of how these two factors combined alter the availability of intact land area for species undergoing elevational range shifts in mountain ranges. We addressed this need by evaluating the elevational distributions of total and intact land area for 1010 global mountain ranges. Here, we adopt the definition of mountain ranges used by the Global Mountain Biodiversity Assessment[25] and use the term 'intact' to refer to areas that are not under intense human pressure that can negatively impact species persistence (see also sections "Results" and "Methods"). We followed existing approaches to classify mountain range topography to one of four mountain topography classes (pyramid, diamond, hourglass, and inverse pyramid) based on the statistical properties of area-elevation distributions for total area[9]. We then reclassified mountain ranges based only on intact area, i.e., areas not under intense human pressure, by

applying a threshold[24] to the human footprint index (HFI), a spatially explicit map of weighted cumulative threat that combines eight separate direct threats from anthropogenic activities: croplands, pastures, roads, railways, navigable waterways, human population density, nighttime lights, and the built environment, circa 2009[26] (Supplementary Fig. 1; see section "Methods"). We then modeled range shifts on all mountain ranges for an extensive set of hypothetical species based on expected temperature changes from a suite of general circulation models (GCMs) under two warming scenarios, assuming species could occupy all available land area in one case and that they would be restricted only to intact land area in a second case. Our approach enabled us to quantify how interactions between topography and current patterns of human pressure potentially influence the amount of intact area available for species following range shifts across the full array of elevations for all the world's mountain ranges.

## Results

**Global patterns of human pressure over elevation.** Human pressure in mountain ranges has resulted in 57% of all mountainous land being considered under intense human pressure (Supplementary Fig. 1). For roughly 24% of ranges (239 of 1010), the entire land area is under intense human pressure (Fig. 1b). The average elevation of peak human pressure in mountain ranges occurred at ~1210 m (range −75 to 6550 m; Supplementary Fig. 2). While human pressure is generally highest at low elevations and declines with elevation, it is not restricted to low elevations: roughly 30% of all land in mountain ranges >4500 m is under intense human pressure (Supplementary Fig. 3). Furthermore, pressure is predominantly focused at the bases of mountains, which can sometimes occur thousands of meters above sea level. For example, the Altiplano in Peru, the Medicine Bow Mountains in the United States, and the Tibetan Plateau all have their bases >2000 m above sea level. Roughly 30% of ranges had peak human pressure within the bottom 5% of their elevational range, with pressure declining rapidly with increasing elevation (Supplementary Fig. 2).

Overall, trends in human pressure over elevation are nonlinear at both global and regional scales (Supplementary Fig. 3). For example, at the global scale, there is a greater proportion of intact land from sea level to 2000 m than from 2000 to 4000 m elevation (Supplementary Fig. 3b). At continental scales, this trend holds for ranges in Africa, Asia, and Oceania, but not for ranges in Europe, North America, or South America, which have equal or greater proportions of intact land at higher elevations (Supplementary Fig. 3c).

**Mountain classification accounting for human pressure.** The frequency and spatial distributions of our mountain topography classifications were consistent with previous classifications of global mountain topography using alternative data sources[9] (Fig. 1a). Roughly 50% of ranges (507 of 1010) were reclassified when calculations were based on the availability of intact land area (Figs. 1b and S4): pyramid mountains accounted for 17% of all mountain ranges, diamond mountains accounted for 30% of ranges, hourglass mountains accounted for 28% of ranges; and inverse pyramid mountains accounted for 2% of ranges. The remaining ~24% of mountain ranges had no remaining intact land area after removing all area under intense human pressure from the analysis and were classified as 'intensified' (Fig. 1b). Reclassifications were geographically heterogeneous; for example, the European Alps changed from originally being classified as a pyramid mountain range to being reclassified as an hourglass mountain range due to disproportionate amounts of human pressure at lower elevations (Fig. 1c: B1 and B2). By contrast, the

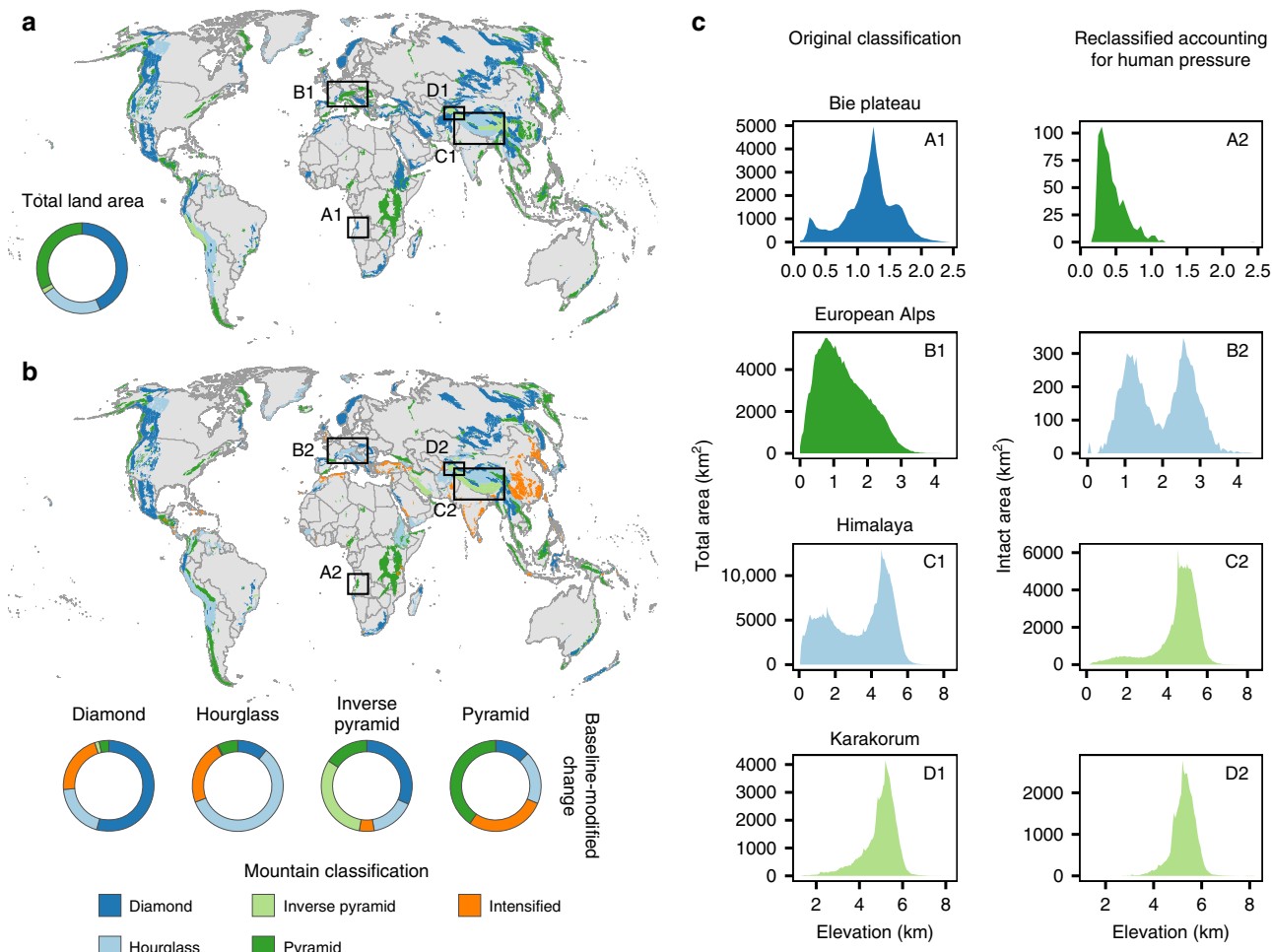

**Fig. 1 The global distribution of topographic classes within mountain ranges.** The distribution of classes when considering all land **a** and intact land **b**. Inset donut plot in **a** shows the proportion of mountain ranges per mountain class considering total land area. Inset donut plots in **b** show the proportion of ranges from their original classification in each mountain class (denoted by donut labels) that were then reclassified considering only intact land area. Mountain ranges classified as intensified have no intact land area remaining when removing all land under intense human pressure. **c** Comparisons of area-elevation distributions considering total land area (left column) and intact land area (right column) for four example mountain ranges, colored by mountain classification. Black rectangles with alphanumeric symbols in **a** and **b** indicate the geographical location of each mountain range in **c**.

Himalayas changed from an hourglass mountain range to an inverse pyramid mountain range (Fig. 1c: C1 and C2). Concentrations of such reclassifications occurred throughout the Great Basin in North America; along the Andes and in mountain ranges of Brazil's Atlantic forest in South America; throughout the Atlas ranges of Morocco, the Ethiopian Highlands, and nearly all mountain ranges of Madagascar in Africa; throughout much of the Middle East and Central Asia, the Indian subcontinent, and mountain ranges of East Asia; throughout much of western and central Europe; and across New Guinea and New Zealand in Oceania (Fig. 1; Supplementary Fig. 4). Mountain ranges that were classified as 'intensified' occurred mainly in northern Africa; the Middle East; and in South, East, and Southeast Asia (Fig. 1b).

**Constraints for species undergoing elevational range shifts.** We assessed how the availability of land area would change for a species shifting its elevational range on each mountain range under two cases—one case where species would occupy all land area within their elevational range, and a second case where species would only occupy intact land area within their elevational range (see examples in Fig. 2). This provides an indication of how more or less vulnerable a species may be to reductions in potential area in each of the mountain ranges in the future under

climate change. We did this by modeling range shifts of a set of hypothetical montane species with a wide variety of elevational range sizes—meant to capture different ecologies, degrees of specialization, and climatic niche breadths—over the complete elevational gradient for all mountain ranges based on mountain range-specific average warming rates across 17 GCMs for two warming scenarios (representative concentration pathways, RCPs 4.5 and 8.5) and on mountain range-specific temperature lapse rates (Supplementary Fig. 5) under these two cases (see "Methods" section). For the purposes of our analysis, our modeled range shifts operate on the assumption that species will closely track shifting isotherms and therefore do not explicitly address lagged responses or disequilibrium dynamics (but see "Discussion" section).

Compared to the initial area of occupancy, the amount of projected area following range shifts was typically greater in the second case where the modeled species were only allowed to occupy intact land area (Fig. 3a). That is, species more often experienced greater percentage of changes in area following range shifts. This was observed when we averaged across all mountain ranges for nearly the entire elevational gradient for all mountain topography classes, and this effect was consistently more pronounced at low elevations

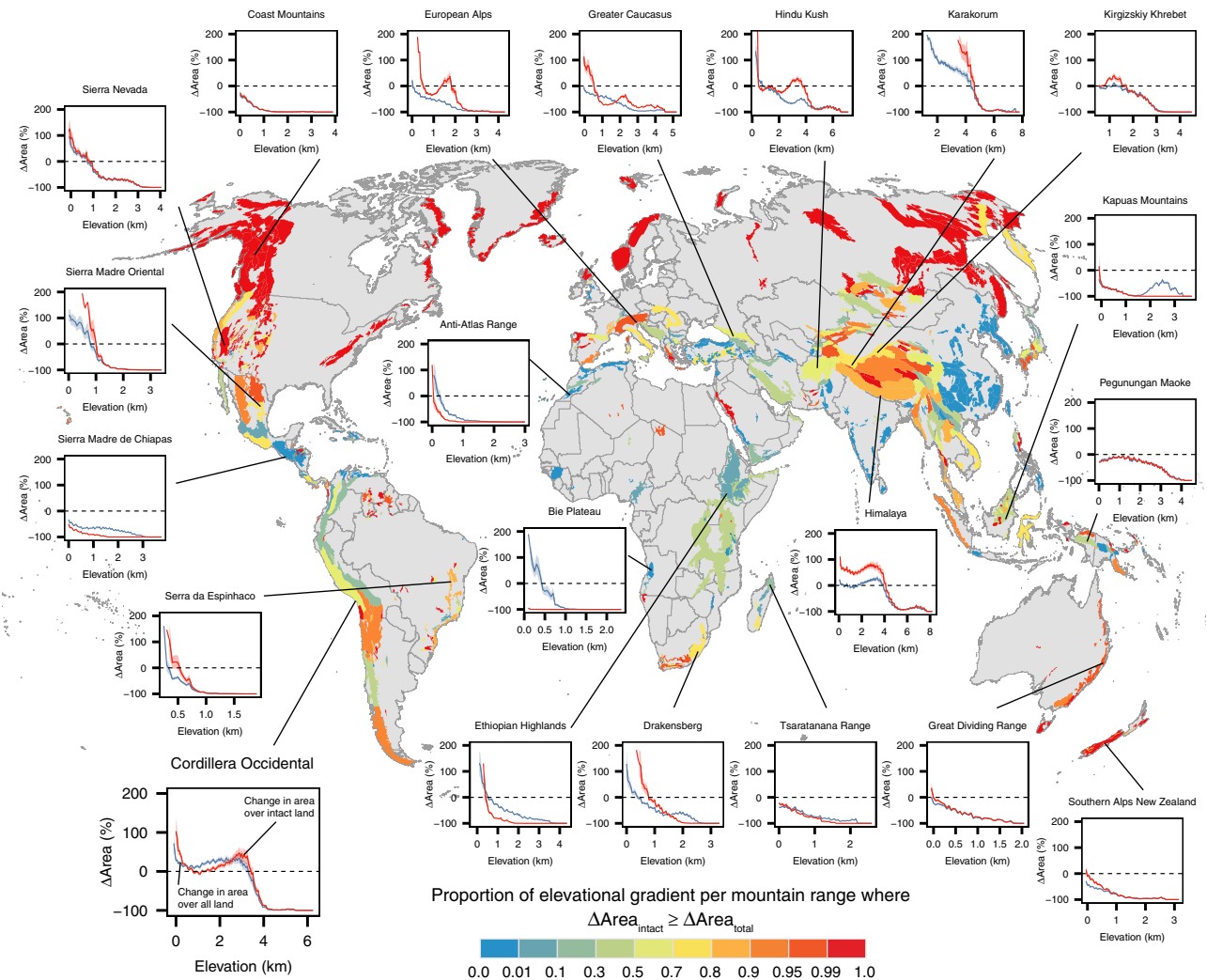

**Fig. 2 Comparing projected area from modeled elevational range shifts with and without accounting for human pressure.** Map of global mountain ranges shows the proportion of the elevational gradient for each range where percentage of change in intact area equals or exceeds the percentage of change in total area following range shifts for a suite of modeled hypothetical montane species. Range shifts are calculated using mountain range-specific rates of mean annual temperature change averaged across 17 GCMs for RCP 8.5 in 2070, mountain range-specific adiabatic lapse rates, and varying elevational range sizes for hypothetical species (see "Methods" section and Supplementary Fig. 9 for descriptive and schematic overviews of the modeling procedure). Insets for select mountain ranges show mean and standard error percentage of changes in area across all modeled species in two cases, one where species are allowed to occupy all land area (blue lines $\Delta\text{Area}_{total}$), and one where species only occupy intact land area (red lines $\Delta\text{Area}_{intact}$). Values of −100% signify that the species have no available total or intact land area remaining and face local extinction. The x-axis elevation represents the lower boundary of elevation prior to the range shift. See the plot for Cordillera Occidental for a detailed legend to other insets. Therefore, the global map depicts the proportion of each mountain range's elevational gradient where the change in area in the second case equals or exceeds the change in area in the first case (i.e., the proportion of elevation where the red curve equals or exceeds the blue curve). See also Supplementary Data 1 for analogous inset plots for all mountain ranges ($n = 1010$) and Supplementary Fig. 7a for the analogous global map using RCP 4.5.

(Fig. 3a). For example, modeled range shifts where species closely track shifting isotherms in the European Alps in the first case led to continued reductions in projected area available for species as total land area tended to decline monotonically with elevation (Fig. 2 inset, blue line). However, in the second case, the modeled range shifts in the European Alps led to area gains up until about 600 m and then again from 1500 to 2500 m, due to shifting away from areas under greater human pressure at lower elevations (Fig. 2 inset, red line). At higher elevations (>2500 m), species encountered area reductions in both cases owing largely to topographic constraints.

However, average responses across all modeled species (i.e., across all elevational range class sizes considered) were variable across mountain ranges and there were some notable exceptions to this general trend described above. For example, in

mountain ranges like the Sierra Madre de Chiapas of Mexico, the Kapuas Mountains on Borneo, the Anti-Atlas Range of Morocco, and the Ethiopian Highlands, current human pressure is expected to reduce land area available for species undergoing range shifts, compounding reductions in area driven by topographic constraints (Fig. 2; see also Supplementary Data 1 for results from all 1010 mountain ranges and Supplementary Software 1 for example code and data for modeling elevational range shifts).

On average, modeled species with smaller elevational range sizes showed larger percentage gains in projected area than species with larger elevational ranges, especially at low elevations (Fig. 4a). Differences were then attenuated above roughly 1500 m on diamond, hourglass, and inverse pyramid mountains, and above roughly 1000 m on pyramid mountains. This was true in both the

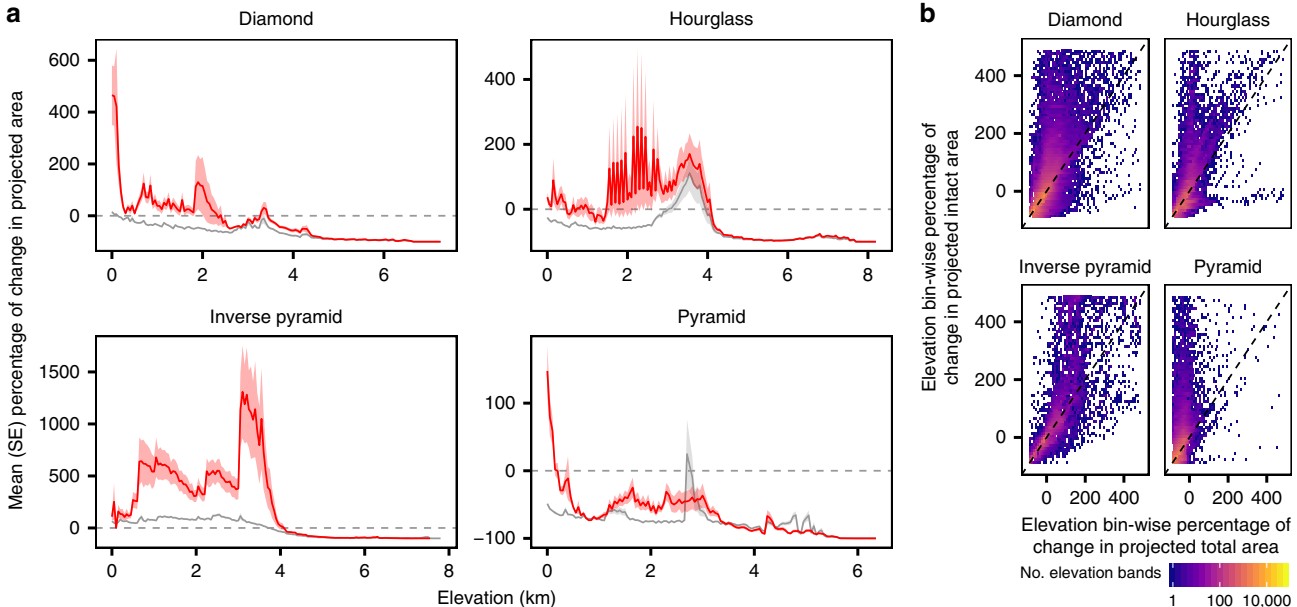

**Fig. 3 Average projected area changes for species undergoing elevational range shifts. a** Mean (lines) and standard error (shaded regions) percentage of change in projected total (gray lines) and intact (red lines) land area across all modeled species and all mountain ranges by mountain classification. **b** Elevation bin-wise comparison of percentage of change in projected total and intact area. Dashed line is 1:1 line. Cells are colored by the total number of elevation bands (50 m) across modeled species and all mountain ranges, with panels arranged by mountain classification based on total area availability. See text and "Methods" section for details of modeled range shifts and Supplementary Fig. 6 for analogous results using RCP 8.5 for panel **a**.

case where species were allowed to occupy all land as well as where species were restricted to intact land only, but the effect was much more pronounced in the latter. This makes sense because as species' elevational range sizes approach the amplitude of a given mountain range, upslope shifts are more likely to result in species losing available area as their upper limits exceed those of mountaintops. We found that the choice of modeled elevational range size influenced the overall proportion of mountain ranges where the percentage changes in area of intact land equaled or exceeded the percentage changes in area of total land (Fig. 4b). In general, species with smaller modeled elevational range sizes tended to decrease this proportion at lower elevations, particularly for hourglass and pyramid mountains (Fig. 4b).

Patterns of mean percentage of changes in projected area following modeled range shifts in the two cases were generally similar under the two warming scenarios we considered (RCPs 4.5 and 8.5; compare Fig. 3 and Supplementary Fig. 6). However, mean percentage gains under RCP 8.5 were significantly greater for diamond and inverse pyramid mountains at lower elevations, and were significantly lower for hourglass and pyramid mountains at lower elevations. We found that the proportion of each mountain range's elevational gradient where the percentage of changes in area of intact land equaled or exceeded the percentage of changes in area of total land following modeled range shifts was generally greater under the RCP 4.5 scenario, though there was significant geographic variation (Fig. 2; Supplementary Fig. 7). In addition, the choice of HFI threshold determining intact land area had little effect on the patterns we observed and was similar across a range of alternate threshold values (Supplementary Fig. 8; see "Methods" section). Our modeling procedure highlights changes in area arising from temperature-induced range shifts that are predominantly upslope, though we acknowledge species respond to other climatic factors and shift heterogeneously along elevational gradients[1,27] (see Supplementary Fig. 9 for a descriptive and schematic overviews of the procedure).

## Discussion

Topography and current patterns of human pressure across the world's mountain ranges influence the extent of intact land area available to species undergoing elevational range shifts. By including topography and a metric of human pressure for all mountain ranges on Earth, we provide a more accurate estimate of species vulnerability to area loss as a result of elevational range shifts under climate change. While human pressure has undoubtedly reduced the amount of habitable area available with potential severe consequences for biodiversity historically[28], we found that human pressure in mountains has functionally changed the 'shape' of mountains when viewed from the perspective of species that are restricted to intact landscapes (Fig. 1). There is evidence from models as well as empirical documentation of species benefitting from climate change by expanding their ranges[27,29] and increasing their population size[30]. Our results suggest that montane species that are restricted to intact landscapes—particularly those at lower elevations—could potentially realize similar benefits following upslope range shifts in at least some portions of the elevational range for the majority of mountain ranges globally (Figs. 2–4).

However, an important caveat is that projections of agriculture[31] and human population dynamics[32] suggest that patterns of human pressure might also show upslope trajectories, akin to those of species responding to warming temperatures. Thus, species shifting upslope might continue to face increasing human pressure over time. Future land-use scenarios hold a high degree of uncertainty and depend on a host of factors, such as rates of agricultural intensification versus expansion[33] and human reliance on existing facilities and infrastructure[34], among others. Moreover, future land-use scenarios do not universally predict increasing human pressure towards higher elevations. For instance, overall agricultural productivity is projected to decline in the tropics due to changes in the timing and length of the growing cycle and significantly reduced potential for multiple cropping[35,36], which may reduce human pressure in tropical

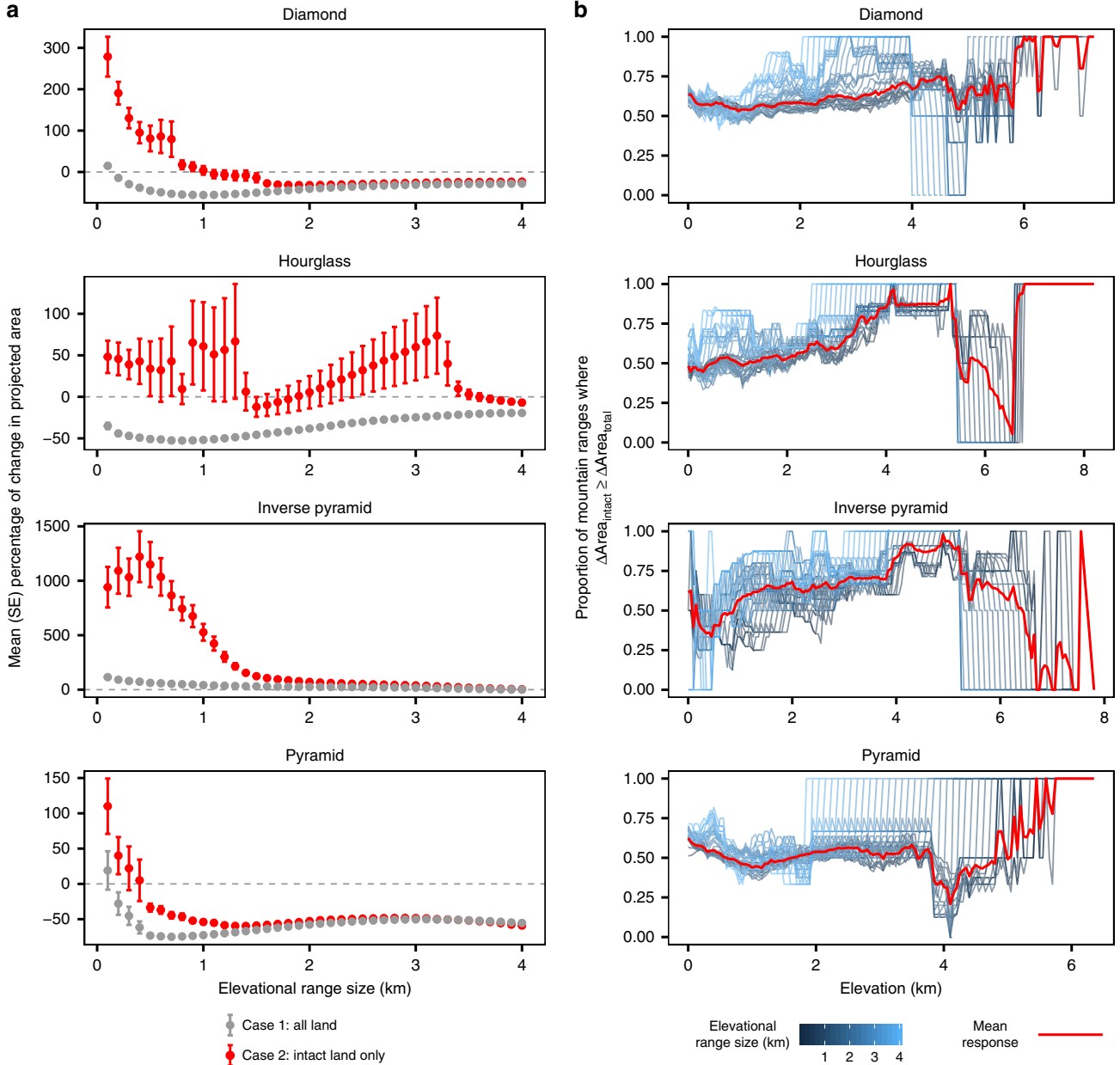

**Fig. 4 The influence of species' elevational range sizes on projected area changes following modeled elevational range shifts. a** Mean and standard error percentage of change in projected total (gray) and intact (red) land area across all modeled species (within elevational range size classes, x-axis) and all mountain ranges by mountain classification. **b** Proportion of mountain ranges where percentage of change in intact land area equals or exceeds percentage of change in total land area following elevational range shifts across the elevational gradient by mountain classification. Blue lines indicate proportions for each elevational range size class considered in modeling (100–4000 m, in 100-m increments); red line indicates the mean response across all range class sizes. See text and "Methods" section for details of modeled range shifts.

mountain ranges over time at all elevations. Higher elevation areas may also offer fewer opportunities for human activities, such as agriculture or road building because they are generally more topographically complex[36]. While we currently lack spatially explicit models of future land-use change scenarios at high enough resolutions in global mountain ranges to incorporate into our vulnerability assessments for species under climate change in this study, future studies would undoubtedly benefit from such information to improve predictions.

While our results counter the existing paradigm that temperature-driven movements necessarily lead to overall area loss for species and may thus increase optimism, it is important to recognize that montane species will face numerous challenges in

accessing and adapting to intact land at higher elevations. Nearly 400 million people permanently live in mountains, and the total number of people jumps to nearly 1.2 billion when including seasonal visitors to mountains[25]. Our analysis revealed that nearly 60% of all mountainous land has been transformed by human activities to a state of intensive human pressure that, in addition to potentially facilitating invasive species and having other indirect effects on ecosystems[37], will likely reduce habitat connectivity and hinder species' dispersal abilities to track climate change[38]. Moreover, while species undergoing range shifts may experience large percentage increases in area in many cases, it is important to note that the total amount of intact area may still be extremely small. For example, in the case of an upslope-shifting

species at the base of the Karakorum Mountains, species may experience a >200% increase in area following a range shift (Fig. 2), but still only have <100 km$^2$ of intact area available to occupy (Fig. 1c). This is an important consideration for all mountain ranges where human activities have reduced the overall area of intact land to nearly nothing, and suggests restoration and rehabilitation activities may still be necessary for long-term sustainability.

We note that our focus on the amount of intact land area—and not its configuration along elevational gradients that could facilitate or hinder connectivity between intact landscapes or the physical environment[39–41]—is a limitation of our study that should be addressed in future work to more accurately understand montane biodiversity responses to global change. In fact, while protected areas are disproportionately biased towards high elevations, comprise more intact landscapes[24], and reduce the rate of intact forest loss from timber harvest[42], they are not evenly distributed or well-connected along elevational gradients in many cases[16]. Consequently, restoring degraded landscapes and strategically planning future protected areas to bolster elevational connectivity will be crucial to increase the chances that species can realize any benefits from increased intact land area at higher elevations.

Indeed, global prioritizations for biodiversity conservation have heavily targeted mountain landscapes for additional protection and restoration. For example, a recent global synthesis identified that the highest global and regional priorities for protected area expansion are located primarily in mountainous regions, including in the Neotropics (Central America and along the Andes and the Brazilian coast), Africa (Madagascar, the Eastern Arc Mountains, and west African forests) and South and Southeast Asia (the Himalayan slopes, Indonesia, Papua New Guinea, and the Philippines)[43]. Where additional protection of intact land cannot meet current conservation targets, restoration priorities have been identified to meet shortfalls in several montane ecoregions, like the Sulaiman Range alpine meadows on the border of Afghanistan and Pakistan, the Madagascar subhumid forests along the Ankaratra Range, and the Huon Peninsula montane rain forests spanning the Finisterre and Saruwaged Ranges in Papua New Guinea[44].

Our results show that protecting and restoring montane landscapes might also act to provide important benefits for lowland biodiversity in mountains under climate change. Rates of plant richness have already increased due to warming-driven upslope range shifts across European mountain ranges[45]. If these patterns are consistent in other mountain landscapes and trends continue, species richness may continue to increase in mountains globally. Some evidence suggests that montane species may be more tolerant of land-use change than lowland species because they evolved in more variable climates and may thus be more adapted to cope with temperature changes arising from habitat modification (e.g., increased temperatures following logging)[46–48]. This could provide further rescue effects from pressure related to human activities.

Related to this issue is whether species can successfully shift their ranges fast enough to keep up with the pace of warming, whether the habitats that species depend on shift in similar directions and with similar magnitudes, and whether species that do colonize higher elevation habitats can persist there. Our analysis assumes species will closely track shifting isotherms, but several studies of plant and animal communities have shown that species range shifts may lag behind shifting isotherms and that such lags influence disequilibrium dynamics between colonization credits and extinction debts[6,49,50]. While there is significant variation in lags across species owing to variation in species' physiological and demographic responses, biotic interactions, and properties of the physical environment[50], some assessments have

reported significant extinction debt is looming for montane species that is more acute for endemic and cold-adapted, high-elevation species[51,52]. Expanding our models to incorporate disequilibrium dynamics, lagged responses, and extinction debt in future work would be an important step to ensuring realistic forecasts of extinction risk for range-shifting species.

There are several ecological, demographic, and political processes that could facilitate landscape recovery in mountainous ecosystems that could potentially provide opportunities for species retreating upslope. Despite global reductions in the area of intact forested landscapes[42], remote sensing of land cover change has indicated that mountain systems across all climate domains have experienced net tree canopy gain and net bare ground loss since the early 1980s[53]. This could represent an improvement in habitat quality and/or quantity for montane species restricted to forests. Similarly, historical and ongoing agricultural land abandonment—the ceasing of agricultural activities on croplands and grasslands due to a host of climatic, environmental, and sociopolitical factors[54]—can in some cases lead to natural regeneration and afforestation[55]. Furthermore, large-scale restoration efforts, such as China's Grain-for-Green Program, have resulted in significant additional forest cover in mountainous landscapes[56], though the net benefits to biodiversity may not equal those provided by natural forests[57].

Overall, our analysis reveals the importance of integrating topography and land use in examining available area for species conservation. Our results strongly suggest that extinction risk from climate-induced range shifts is highly context dependent in mountain ranges and can be driven in large part not only by the 'fundamental shape' of the mountain, but its 'realized shape' after accounting for human pressure. Indeed, conservation actions informed by analyses of topography alone may be misleading. For example, restoration actions might be targeted towards low elevations in the Cordillera Oriental in Colombia, a diamond mountain, when based on topographic patterns alone. However, factoring in human pressure reveals that the mountain range is shaped like a pyramid for species restricted to intact landscapes, which means conservation actions may better be targeted towards mid and high elevations where species may have limited access to intact land area when shifting upslope.

More broadly, analyzing global topography and land use in concert reveals that species driven upslope in response to future warming may have access to more intact land away from intense human pressures that many montane species currently face. Conservation practitioners must integrate topography and human pressure to accurately assess species vulnerability to climate change.

## Methods

**Mountain ranges**. We obtained a previously published global dataset delineating 1010 mountain ranges that account for 26 million km$^2$ of terrestrial land (~17.5% by area) and 83.7% of the Earth's mountainous terrain[25]. Delineations in the dataset were made by expert evaluation using maps, atlases, and inventories of mountain ranges, and boundaries were subsequently informed by terrain ruggedness, which is a measure of the elevational change between focal and neighboring cells of a digital elevation model (DEM). Boundary delineations were optimized to maximize the inclusion of rugged terrain while minimizing nonrugged terrain. To our knowledge, it is the most comprehensive dataset on mountain ranges with the greatest number of mountain ranges delineated currently available.

**Elevation data**. We obtained a high-resolution, near-global, void-filled DEM from the NASA Shuttle Radar Topography Mission (SRTM) at 3 arc-second (90 m) resolution (SRTM3). We then resampled the SRTM3 raster to the 1 km$^2$ resolution of the HFI raster using bilinear interpolation. The SRTM3 DEM extends from approximately 56°S to 60°N. Sixty mountain ranges extend beyond 60°N, so we combined the resampled SRTM3 DEM with a second DEM at 30 arc-second (1 km$^2$) resolution (SRTM30), which already matched the resolution of the HFI raster. This resulted in a seamless elevation dataset that matched the spatial resolution of the HFI (see below). All elevation data are available at https://lpdaac.usgs.gov.

**Human pressure**. We obtained a previously published global map of the HFI representing a metric of human pressure[23]. The HFI is a weighted cumulative threat map at a spatial resolution of 30 arc-seconds (1 km$^2$) that combines eight separate direct threats from anthropogenic activities, circa 2009: croplands, pastures, roads, railways, navigable waterways, human population density, nighttime lights, and the built environment. Global maps of each threat were scaled by the original authors based on their degree of influence on the terrestrial environment: the built environment was given a value of 0 or 10 (all areas mapped as built given score of 10); human population density and nighttime lights were scaled to a continuous range of 0–10; croplands were given a value of 0 or 7 (all areas mapped as crops given score of 7); pasture was given a value of 0 or 4 (all areas mapped as pasture given score of 4); roads were scaled from 0 to 8; railways were given a value of 0 or 8 (all areas within 500 m of a railway given a score of 8); and navigable waterways were scaled to a range of 0–4[26]. Scores for individual threats were then summed and weighted by the original authors to make a composite map with a range of 0–50. HFI values were then extensively validated against satellite imagery[26] yielding high confidence they represent conditions of human pressure.

**Classifying mountain topography based on total and intact land area**. For each mountain range we calculated the skew and modality of the extracted elevation values. We followed established methods[9] to assign mountain ranges to one of four classes based on combinations of skew and modality. We assigned mountain ranges with a dip value (a test statistic that measures the degree of multimodality of an empirical distribution, calculated using Hartigan's dip test[58]) > 0.01 and with significant ($p < 0.05$) deviations from unimodality to the hourglass classification (irrespective of skew). For all other mountain ranges, we assigned those with Type-I skewness[59] $\geq 0.5$ to the pyramid classification, those with skewness $\leq -0.5$ to the inverse pyramid classification, and the remainder to the diamond classification (Fig. 1a). We then repeated this procedure after removing all elevation values that were derived from pixels under intense human pressure based on the HFI to assign each mountain a classification based on its distribution of intact land area over elevation. To do this, we followed previous analyses using the HFI to set a threshold that separates intact land from land under intense human pressure[24,60,61]. Consequently, we used a threshold value of ≥4—roughly equivalent to land being converted to (at least) pasture and reasonably approximating when human activities have converted land to a non-natural state—to indicate land area under intense human pressure, such that HFI values < 4 were considered intact (Supplementary Fig. 1; see also 'Sensitivity analyses' below for treatments using alternate thresholds). For those mountain ranges where all elevations were found to be under intense human pressure using our threshold, we assigned them to an 'intensified' classification (Fig. 1b).

**Human pressure over elevation**. We clipped both global HFI and DEM rasters to the boundaries of the mountain range delineations and extracted the raster values from each layer for each mountain range. We created histograms of the elevation data binned within 50-m elevational bands—chosen to produce accurate elevation–area relationships while aligning with the spatial scale of empirically documented elevational range shifts on decadal-to-century time scales[4,62]—to visualize trends in total land area over elevation for (i) each mountain range, (ii) for all mountain ranges within continents, and (iii) for all mountain ranges combined.

To visualize trends in intact land area over elevation, we created a second set of histograms of the elevation data comprising only intact land using the threshold as defined above, again binned within 50-m elevational bands. For each range we also calculated the proportion of land area intact within each elevational band as the number of HFI values <4 in the band divided by the total number of raster values in the band (Supplementary Fig. 3). We also used this approach to calculate the overall proportion of intact land area for each entire mountain range (i.e., across all elevational bands).

We assessed bias in human pressure over elevation in two ways. For each mountain range, we first calculated the elevation of peak human pressure. We did this by determining the elevational band where the proportion of intact land area was minimized. In cases where multiple elevational bands had equal proportions, we took the median value. We next calculated the relative elevational position of peak human pressure per mountain range by determining the position of the elevational band denoting peak human pressure relative to each mountain range's amplitude. This value was then scaled from 0 (a mountain range's base) to 1 (a mountain range's peak; Supplementary Fig. 2). For example, two mountain ranges with peak human pressures at 1000 m where the first mountain ranges from sea level to 2000 m and the second mountain ranges from 1000 to 3000 m would receive values 0.5 (i.e., halfway up the elevational gradient) and 0, respectively.

**Modeled range shifts**. We modeled range shifts for a suite of hypothetical montane species on each mountain range in two cases—one with and one without removing land area under intense human pressure. To do this, we created a series of hypothetical species to subject to temperature-induced range shifts over all mountain ranges in the world. We first generated species with a wide variety of elevational range sizes, from 100 m up to the total amplitude of a given mountain range, in 100-m increments. For mountain ranges with amplitudes >4000 m, we set the maximum elevational range size to 4000 m. We then

distributed each species within the suite such that its lower elevational range started every 50-m of elevation, starting at the base of the mountain. When species placements would result in the species upper elevational range extending beyond the elevational gradient of the given mountain range, we removed that species from the suite. This resulted in a species set per mountain range consisting of all possible combinations of species elevational range sizes (in 100-m increments) starting at all possible elevational bands (every 50-m of elevation) where species elevational ranges are completely contained within the elevational gradient. Consequently, the number of species modeled, $s$, varied by mountain range and was calculated by

$$s_i = b_i \times (a_i/100)/2 \text{ for } a_i \leq 4000 \text{ m when } b_i \text{ is even,} \quad (1)$$

$$s_i = (b_i - 1) \times ((a_i + 50)/100)/2 \text{ for } a_i \leq 4000 \text{ m when } b_i \text{ is odd,} \quad (2)$$

$$s_i = 1600 + (b_i - 80) \times 40 \text{ for } a_i > 4000 \text{ m when } b_i \text{ is even, and} \quad (3)$$

$$s_i = 1600 + (b_i - 80.5) \times 40 \text{ for } a_i > 4000 \text{ m and when } b_i \text{ is odd,} \quad (4)$$

where $b$ is the number of 50-m elevational bands and $a$ is the amplitude (maximum minus minimum elevation) of mountain range $i$. For example, a mountain range extending from 1000 to 4000 m would have $b = 60$ elevational bands and $a = 3000$ m amplitude for a total of $(60 \times (3000/100))/2 = 900$ species modeled (a schematic and further details are presented in Supplementary Fig. 9). We then subjected each species to an elevational range shift on each mountain range depending on mountain range-specific warming scenarios and adiabatic lapse rates.

We calculated rates of warming separately for each mountain range to capture geographic variation in warming across mountains globally[63]. To do this, we extracted projected mean annual temperature data from within each mountain range for 17 GCMs (ACCESS1-0, BCC-CSM1-1, CCSM4, CNRM-CM5, GFDL-CM3, GISS-E2-R, HadGEM2-AO, HadGEM2-CC, HadGEM2-ES, INMCM4, IPSL-CM5A-LR, MIROC-ESM-CHEM, MIROC-ESM, MIROC5, MPI-ESM-LR, MRI-CGCM3, and NorESM1-M) from two representative concentration pathways (RCPs 4.5 and 8.5) for 2070 (average for 2061–2080) from WorldClim v1.4[64]. We calculated the pixel-wise temperature difference between each projected temperature layer and current mean annual temperature, also using data from WorldClim v1.4 to enable unbiased comparisons with the future projections. We then averaged all resultng difference rasters and took the mean value across all pixels within each mountain range to determine mountain range-specific average rate of warming for each RCP (Supplementary Fig. 5b, c).

We calculated adiabatic lapse rates separately for each mountain range to capture geographic variation in temperature–elevation relationships arising from complex topographic and orographic features in mountains[65]. To do this, we extracted mean annual temperature data from within each mountain range using the current (1970–2000) climate data from WorldClim v2.0[66]. We then fit linear models of temperature–elevation and used the slope of this relationship (the coefficient) as the lapse rate for each mountain range (Supplementary Fig. 5a; Supplementary Data 2).

For both cases (one with and one without removing land area under intense human pressure) and for each hypothetical species modeled, we calculated the amount of area within the range of the species prior to the range shift (Area$_{baseline}$) and after the range shift (Area$_{projected}$). We then calculated the percentage of change in area for a given hypothetical species starting from a given 50-m elevational band as

$$\% \text{ of change in area} = ((\text{Area}_{projected}/\text{Area}_{baseline}) - 1) \times 100 \quad (5)$$

The results for the first case reflect a situation where a species can occupy any area regardless of human pressure. By contrast, the second case relies on baseline and projected areas only from intact land area and therefore reflects a situation where a species can only occupy intact area where HFI values were <4. We contrasted the results of the two cases by separately plotting the average percentage of change in projected total area ($\Delta$Area$_{total}$) and the average percentage change in intact land area ($\Delta$Area$_{intact}$) across all hypothetical species over elevation, separately for each mountain range and also by mountain classification. We provide several detailed examples of this procedure as insets in Fig. 2 and provide the full set of plots for all mountain ranges in Supplementary Data 1. Calculating the mean and standard error percentage of change in projected total and intact land area across all species separately per elevational band (Fig. 3a), and creating a heatmap of expected changes in total versus intact land area at the scale of individual elevational bands across all species (Fig. 3b), provides an assessment of the average response to elevational range shifts under the two cases considered. For descriptive and schematic overviews of our modeling procedure, we refer readers to Supplementary Fig. 9.

**Sensitivity analyses**. We performed three sensitivity analyses to test whether and how our results would be influenced by the choice of (i) the HFI threshold value used to designate intact land area, (ii) the warming scenario (RCP) considered, and (iii) the elevational range size of a hypothetical montane species used in modeling range shifts.

For our first sensitivity analysis, to test how our results were influenced by the choice of the HFI threshold used to designate intact land area, we repeated our analyses after using a more stringent (HFI value ≥ 3) and a more conservative (HFI value ≥ 7) threshold for intact land area following an existing methodology[24]. HFI values < 3 roughly denote land that has not been developed to the level of a pasture and contains no heavy infrastructure, while HFI values < 7 roughly denote land that has not been converted to cropland. Using these alternate thresholds, we repeated the mountain classification, human pressure, and modeled range shift analyses described above, using warming scenarios derived from RCP 8.5. The results using these two alternate thresholds were qualitatively similar to those obtained using our original threshold value of 4 (Supplementary Fig. 8). Generally speaking, using a threshold value of 7 resulted in a pattern more similar to that using no threshold value (i.e., the total land case), while using a threshold value of 3 resulted in a pattern where changes in projected area for species undergoing range shifts were slightly greater than those using the threshold value of 4. The largest discrepancies in results driven by the choice of threshold values were observed for inverse pyramid mountains when considering a threshold value of 7.

For our second sensitivity analysis, to test how our results were influenced by the choice of the warming scenario used to model elevational range shifts, we repeated our analyses using RCP 4.5, which represents a more moderate warming scenario compared to RCP 8.5, again using projected mean annual temperature data using WorldClim v1.4[64]. On average, mountain-range specific warming rates using RCP 4.5 were two-thirds as high as when using RCP 8.5 (Supplementary Fig. 5b, c). This led to smaller modeled elevational range shifts for most mountain ranges. Patterns of mean and standard error percentage of change in projected area assessments under RCP 4.5 were qualitatively similar to those under RCP 8.5 at the global scale (Supplementary Fig. 10), but mean percentage gains under RCP 4.5 were significantly lower for diamond and inverse pyramid mountains at lower elevations, and were significantly greater for hourglass and pyramid mountains at lower elevations (Fig. 3; Supplementary Fig. 6). The proportion of each mountain range's elevational gradient where the percentage of change in area of intact land equaled or exceeded the percentage of change in area of total land following modeled range shifts was generally greater under the RCP 4.5 scenario, though not always (Fig. 2; Supplementary Fig. 7).

Finally, for our third sensitivity analysis, to test how our results were influenced by the choice of the elevational range size of a hypothetical montane species used in modeling range shifts, we considered a wide variety of elevational range sizes from 100 m to a maximum of 4000 m meant to represent species with different ecologies, degrees of specialization, and climatic niche breadths in our modeled range shift procedure. We then plotted mean and standard error percentage of change in projected area following range shifts for all modeled species within elevational range size classes across all mountain ranges by mountain classification to assess the influence of elevational range size (Fig. 4a). Across all mountain classes, smaller elevational range sizes were associated with larger mean percentage of change in projected area gains. Gains were generally similar for species with elevational range sizes >1500 m on diamond, hourglass, and inverse pyramid mountains, and for species with elevational range sizes >1000 m on pyramid mountains.

We also investigated how the choice of elevational range size affected our calculation of the proportion of mountain ranges where the percentage of change in area of intact land equaled or exceeded the percentage of change in area of total land following modeled range shifts. In general, patterns were qualitatively similar across elevational range sizes on diamond, hourglass, and pyramid mountains. We found a trend for greater elevational range sizes to be associated with greater proportions for inverse pyramid mountains, and this trend was also slightly apparent for hourglass mountains. However, generally speaking, the overall average response (i.e., response averaged over all elevational range sizes) was similar to that of a given elevational range size.

**Reporting summary**. Further information on research design is available in the Nature Research Reporting Summary linked to this article.

## Data availability
The SRTM elevation data are freely available at https://lpdaac.usgs.gov. The Human Footprint index data are freely available from the original author's Dryad Digital Repository (https://doi.org/10.5061/dryad.052q5). The WorldClim current and future mean annual temperature data are freely available at http://worldclim.org. The mountain range boundary delineations from the Global Mountain Biodiversity Assessment are freely available at https://ilias.unibe.ch/goto_ilias3_unibe_cat_1000515.html.

## Code availability
R scripts for performing the mountain classification analysis and modeling elevational range shifts for an example mountain range are available as Supplementary Software.

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

## Acknowledgements

P.R.E. was supported by the David H. Smith Conservation Research Fellowship administered by the Society for Conservation Biology and funded by the Cedar Tree Foundation. We thank the Berkeley Research Computing for guidance and use of the supercomputing cluster for portions of the analysis.

## Author contributions

P.R.E. conceived and designed the study, performed the analysis, and drafted the manuscript. W.B.M. and A.M.M. contributed ideas and feedback on the analysis and edited the manuscript.

## Competing interests

The authors declare no competing interests.
