## [Peer Review File · Nature Communications]

Reviewers' Comments:

Reviewer #1:

Remarks to the Author:

Manuscript review: Topography and human pressure in mountain ranges alter expected species responses to climate change

Recommendation: Minor revision

Summary: This paper covers an important aspect of species responses to global warming in montane regions by analyzing changes in area with elevation in mountain ranges of different geometric configurations. The overall topic should be interest to ecologists, conservation biologists, and some earth scientists.

The analysis combines changes in area along with the Human Footprint metric of human impacts on land use and arrives at two important conclusions. First is that the hypsometric shape of mountain ranges differs from the shape when constrained by the area of "intact" land area by a measure of human pressure. Second is that, in some mountain ranges, species moving upslope will encounter more "intact" land area than at lower elevation, thereby gaining potential habitat. These are worthwhile insights. However, the authors could strengthen the paper by recognizing some of the complexities that are glossed over or receive no mention at all. Also, some of the phrasing could be usefully clarified.

Explain in more detail what "human pressures" involve; which of these have the greatest impact on montane biodiversity?

The phrase "intact land area" is misleading. "Intact" has multiple meanings, one being that a specific parcel of land retains most or all of its native ecosystem diversity and functions; another is that land is spatially connected (e.g., the opposite of fragmented). However, even under stable climates, similar habitats in montane regions are fragmented and sometimes widely separated by the topography itself. This second meaning of intact does not apply well to montane regions. Some other term would be useful to avoid ambiguity.

The authors do not mention habitat connectivity until the discussion; how much "intact" land area is accessible at any elevation is not part of the analysis. The authors would do well to acknowledge this topic as a limitation of their study.

Explain how "mountain range" is defined in this study, even if borrowed from Korner's previous analysis. This is important because 1010 mountain ranges seems like a lot; other recent studies of global mountain ranges on the continents list 200-300 ranges by different criteria.

Comments on specific passages:

L 40: elaborate on the term "detailed topography." What aspects of topography are included, which are excluded?

L 45: Explain what is meant by "human pressure." Which activities have the greatest negative impact on native biodiversity? This topic needs elaboration within the main text.

L 52: Explain the meaning of "intact" here (if authors keep this term). Must the land area under consideration be continuous (connected) in order to be habitable range? What proportion of montane

species survive successfully as metapopulations?

L 56: Explain what "mountain range" means in this study.

L 58: Mention here what model of climate change is used as the basis for modeling upslope range shifts. Somewhere it is worth noting that temperature is not the only important aspect of global climate change; changes in precipitation and orographic gradients are quite significant in montane regions. In some studies (e.g., Moritz et al. 2008, *Science* v 322), mammal species shift in different directions (up, down, not at all).

L 87 and elsewhere: "predominately" should be "predominantly."

L 90-93: explain here the different kinds of human pressure and activities and the gradient of pressure represented by your scoring system.

L 100-104: Run-on sentence.

L 115-117: this passage needs rephrasing. Since the study does not include a consideration of relevant habitats (or their connectivity) for any species, then it cannot assess vulnerability to habitat loss. Discuss in terms of land area available for potential habitat.

L 127-128: This result needs clarification. The amount of "intact" land area must be a subset of the total land area. Thus, how can the second scenario result in a greater amount of projected "habitable" area? Explain more clearly that your results are about proportional change rather than absolute area.

L 188-191: The logic of this passage is unclear. There is already a lot of agricultural activity at high elevations in tropical regions. Why would this not continue or even expand under global warming, especially as low-elevation regions become too warm or dry for successful farming?

L 204: Habitat connectivity is mentioned here in the discussion. It seems like a crucial aspect of land area for successful range shifts.

L 229: Why would ecological communities in montane regions be more tolerant of land-use change than those at low elevations? Also, correct the phrasing in this sentence: "ecological communities" are equated to "species" but these are not the same thing.

L 246-248: The last sentence of main text has unclear logic. On the one hand human population and activities are likely to increase at higher elevations; on the other hand, there is greater protection for montane ecosystems than in lowlands. It's not clear though which of these trends will have the greater impact on native biodiversity.

Methods section

L 251-255: Explain here are mountain ranges are recognized.

L 272-273: What does the language in parentheses mean? Explain for the reader who has not read the original Human Footprint report.

L 281: What is a dip value?

L 294: The "intensified" category needs explanation in the figure caption for Figure 1.

L 306, 308: proportion OF land area

L 328-329: the assumption of a fixed adiabatic temperature lapse rate is questionable, since the actual lapse rate varies around the world, especially around mountain ranges. The moist adiabatic lapse rate varies in relation with temperature (so with latitude and elevation).

Figure 1: The area v elevation map for the Greater Caucasus looks like pyramid mountain range, even if it meets the skewness criterion for the diamond shape. Can you choose a better example? Make sure to explain what "intensified" means in the figure caption.

Figure S1: It's unclear how the maps in (a) and (b) relate to each other. If the Yellow for intensified is from the same dataset in both maps, then why do so few areas show up as yellow in (a)?

Catherine Badgley, University of Michigan

Reviewer #2:

Remarks to the Author:

General comments to the Editorial Board and to the authors

By focusing on mountain ecosystems worldwide, the authors aim at assessing the impact of species redistribution induced by global warming while accounting for the effects of both human pressures on the environment and the shape of each mountain range on the amount of land area that will remain available for mountainous species along the elevational gradient. This is a very important research question which has important implications for the future of mountain biodiversity. Yet, I do have serious concerns regarding the analytical framework proposed by the authors to answer this timely question. I do not have strong concerns regarding the first step of the authors' analytical framework which focuses on assigning each mountain range to a given "shape" based on either total land area or "intact" land area. My main concern is about the second step of the authors' analytical framework, when the authors rely on a single hypothetical species to assess the potential change in available "intact" area, following climate warming and isotherms' shifts along the elevational gradient, throughout all mountain ranges worldwide. This is a very strong and problematic, in my opinion, simplification of the reality. First of all, I would recommend the authors to use a multi-species approach to assess the fate of mountain biodiversity in terms of loss or gain in the available land area under future climate change. Second, I would recommend the authors to rely on species response curves along the temperature gradient rather than relying on a single species' elevational range of 800-m range whatever the mountain range. Indeed, following the authors' approach, this means that the hypothetical species may have different temperature and elevational margins from one mountain range to the other. Thus it is difficult to interpret the main findings. I would rather recommend the authors to use a set of virtual species (see Leroy et al., 2016) to create several ($n = 1000$, for instance) species response curves along the temperature gradient and then assess the overall (mean and standard deviation) change in available land area following a given climate change scenario (see WorldClim or CHELSA data for future climate change scenarios) and the subsequent shift of each of the virtual species. Using such an approach should provide much more realistic and general findings than the current findings based on a single hypothetical species: the sample size is equal to one here!

Besides, I would also warn the authors about the implications of assuming a synchronous biotic response of species to climate warming based on a $+2^{\circ}\text{C}$ warming rate and a fixed temperature lapse rate of $-6.2^{\circ}\text{C}/\text{km}$. Indeed, species are unlikely to shift as fast as the isotherms are shifting, even in

mountain ecosystems. There are many clear reasons to believe that living organisms will lag behind the shifting isotherms (Alexander et al. 2018). In fact, the velocity of species range shifts for terrestrial organisms, even in mountain ecosystems, is clearly lagging behind the velocity of isotherm shifts (Lenoir et al. 2019). Thus, these kind of disequilibrium dynamics should be discussed by the authors.

Finally, I do have several more minor comments that I have listed below. I hope these comments and suggestions together with my main recommendations will help the authors improve the quality of their research.

Specific comments to the authors

Lines 17-18: Maybe also refer to more specific research papers (not review or meta-analyses papers) that did show with hard data that animal (Moritz et al., 2008) and plant (Lenoir et al., 2008) species are shifting their range upward in mountain ecosystems as climate warms. It is fine to cite review papers and I also like Chen et al. (2011) paper, but it is also very important to acknowledge the original research papers that are at the basis of such quantitative reviews or meta-analyses. Besides, regarding review papers on climate-related range shifts, note that there are more recent ones focusing on both terrestrial and marine systems (Lenoir & Svenning, 2015; Lenoir et al., 2019).

Lines 26-28: Indeed, this is a very important point. Please see Graae et al. (2018) for a paper discussing the implications of fine-grained topography and short-distance escapes for species survival in mountain systems as climate warms.

Lines 50-53: Yes, indeed, there is a clear research gap here. Yet, you may refer to Guo et al. (2018) here for an attempt to account for both climate change and land use in mountain systems on our understanding of species range shifts in response to global warming.

Line 58: Not clear here what you mean by "species" and on which species you are focusing on to model upslope range shifts. Only in the results' section the reader realizes that you are focusing on a single ($n = 1$) hypothetical species which elevational range is 800 m wide. This is a bit hard to trust and believe on the generalization of your findings which are based on such a strong simplification of the reality. Like I mentioned in my major concerns, you should really consider a multi-species approach for more realistic results and to also account for the uncertainty around your "single" simulation.

Lines 69-88: This very first paragraph of the result section is not bringing new findings. I would rather report these findings in the supplementary information as this is more or less a repetition of what has already been published in the scientific literature (cf. Elsen & Tingley, 2015). The new findings only start from line 89.

Line 89: Not clear what "intact" land area means here. I also assume that "intact" land area for a given mountain range may differ in land use to what is considered "intact" land area in another mountain range, right? If that is the case, isn't it problematic? Would it not be better to focus on a given land use type and assess how the available surface area will change for that focal land use type (e.g. forests)? By doing so, it would be easier to interpret the overall finding/pattern across all mountain ranges worldwide. Here, you are likely to mix very different land uses that are considered "intact" across the different mountain ranges.

Lines 98-100: Is it a new type of mountain "shape" here?

Lines 113-146: Like I wrote in my general comments, I think that using a single hypothetical species here is a very strong and wrong simplification of the reality which may potentially lead to biased results. Indeed, such a simulation does not account for the new species shifting upslope from the lowlands. Your main finding and thus the main message of the paper is clearly biased by the fact that you consider a single species which elevational range size is 800 m wide. Do you really think that we can then generalize your main finding to all mountain species based on such a strong simplification?

Line 120: Here, you wrote that each hypothetical species is distributed along a 800-m wide elevational gradient starting from the mountain range's base. This means that from one mountain range to the other, the hypothetical species has different elevational and temperature range limits, right? How does that affect your results? That is why I strongly recommend to use a multi-species approach and also a virtual species approach by building several virtual species based on thermal response curves (cf. the species niche) rather than using elevational response curves. Indeed, 1000 m a.s.l. within one mountain range in the Tropics does not mean the same for another mountain range located at higher latitudes. That is why it would be safer to rely on species response curves along the temperature gradient rather than relying on fixed species elevational range of 800 m.

Lines 121-122: This is another very strong assumption to use a fixed temperature lapse rate across all mountain ranges. Besides, the reference cited here (#20) is not a very good one or relevant one to backup such a choice of focusing on a single value for the adiabatic lapse rate. There are several studies suggesting that the temperature lapse rate may vary a lot from one mountain range to the other (see references cited in Minder & Lundquist, 2010). These two authors actually wrote in their paper: "A number of observational studies using networks of temperature sensors have revealed that spatially uniform and temporally constant lapse rates of 6–6.5°C km⁻¹ are not representative of actual surface conditions over the Appalachian mountains [Bolstad et al., 1998], the European Alps [Rolland, 2003], the Qinling Mountains of China [Tang and Fang, 2006], the central Rocky Mountains [Blandford et al., 2008], and Arctic glaciers [Gardner et al., 2009]. In all these regions mean surface lapse rates differ appreciably from the often-used 6–6.5°C km⁻¹ values". Hence, it is clear that using a single adiabatic lapse rate for all mountain ranges worldwide is another very strong simplification of the reality, which makes a lot of simplifications... That is another reason why I would strongly recommend you to use species-specific response curves along the temperature gradient rather than relying on a single fixed species elevational range and adiabatic lapse rate to estimate the future shift of mountainous species along the elevational gradient. Using temperature data under current and future conditions will allow to circumvent this issue of having to decide on a given fixed value for the adiabatic lapse rate. Besides, I would also be careful about the implications of assuming a synchronous biotic response of species to climate warming. Indeed, species are unlikely to shift as fast as the isotherms are shifting, even in mountain ecosystems. There are many reasons to believe that living organisms will lag behind the shifting isotherms (Alexander et al. 2018). In fact, the velocity of species range shifts for terrestrial organisms, even in mountain ecosystems, is clearly lagging behind the velocity of isotherm shifts (Lenoir et al. 2019). This should be acknowledged or at least discussed in the discussion section of the paper.

Lines 141-143: Ok, but what about the choice of the range limits along the elevational gradient? Did you run a sensitivity analysis for that as well?

Like 146: You only tested for a larger elevational range size (1600 m), right? Not for a smaller elevational range size since the elevational range size reported in the paper for the hypothetical species is already 800 m!!! Sorry, but I am confused here.

I really enjoyed reviewing your work and I sincerely hope that my comments will be helpful to you to improve the quality of your research.

Cheers,

Jonathan Lenoir

Suggested references

Alexander et al. (2018) Lags in the response of mountain plant communities to climate change. *Global Change Biology*, 24: 563-579

Graae et al. (2018) Stay or go – how topographic complexity influences alpine plant population and community responses to climate change. *Perspectives in Plant Ecology Evolution and Systematics*, 30: 41-50

Leroy, B. et al. 2016. virtualspecies, an R package to generate virtual species distributions. *Ecography*, 39: 599-607

Lenoir et al. (2008) A significant upward shift in plant species optimum elevation during the 20th century. *Science*, 320: 1768-1771

Lenoir & Svenning (2015) Climate-related range shifts – a global multidimensional synthesis and new research directions. *Ecography*, 38: 15-28

Lenoir et al. (2019) Species better track the shifting isotherms in the oceans than on lands. *bioRxiv*, <https://doi.org/10.1101/765776>

Minder & Lundquist (2010) Surface temperature lapse rates over complex terrain: Lessons from the Cascade Mountains. *Journal of Geophysical Research*, 115: Issue D14

Moritz et al. (2008) Impact of a Century of Climate Change on Small-Mammal Communities in Yosemite National Park, USA. *Science*, 322: 261-264

Reviewer #3:

Remarks to the Author:

The paper quantifies the “true” available surface area as species range shifts move upslope along the elevational gradient in mountains due to climate warming. The available surface area is naturally affected by humans (here quantified by the Human Footprint Index) and thus the authors compare the total potential available surface area (if the entire mountain would have intact land area) vs what is truly available when human presence and thus land use are included. They use a similar approach taken by Elsen and Tingley (2015) in classifying the shape of mountains around the world, but now using the recently published dataset of mountains comprising of an inventory of all mountain ranges globally (n = 1010). They first test if mountains change their ‘Elsen and Tingley’-shape by comparing the potential (total area) vs intact surface area ([total area] – [area used by humans]), and describe in more detail an overview of mountains globally that are under high human pressure. To demonstrate the implications of this change in surface area, the authors then model how upwards shifts of taxa would be affected by the available surface area along the elevational gradient under a 2 degrees warming scenario. They do so by modelling two types of hypothetical montane vertebrate taxa, those with a relatively short elevational range (800 vertical meters) and a wide elevational range (1800 vertical meters). The main results of the authors focus on the percentage change in projected total

and intact land area for 50-m elevational bands for all mountains globally. The authors also show at which relative elevation the peak in human pressure occurs. Summarizing, their results show the intensity of intact area loss along lower elevations and the importance of taking into the account the topography of mountains and the true available surface area that can potentially be used by taxa as they move upwards under warming climate conditions.

Overall, I enjoyed very much reading this manuscript and thinking about the implications presented by the figures. The concept of total vs intact area is very intuitive and it is a very important shout out to the numerous modelling studies of montane taxa that only take into account the total potential surface area, but not what area can actually be occupied when range shifts occur. It is so intuitive that many researchers would be surprised (and perhaps frustrated) that they did not think about it before, setting a very solid ground for this manuscript to attract a high number of citations.

I think the overall results of this manuscript provides a very useful reference on the consequences of human pressure in mountains and the need to think more "3D" in terms of mountain conservation management, but here I would like to follow up on the authors' Discussion section which is much focused on making the link to conservation and the overall implications of the results.

The emphasis of the paper and figures concentrates on comparing surface area in [percentage] values and I feel that that obscures a bit the emergency state of the remaining intact surface area along low and mid-elevation areas, and also the seemingly high number of mountains where surface area change is increasingly negative moving upslope (Fig. 3a). Most benefit of moving upslope seem to occur up to 1000 masl (Fig. 2) and for mountains with Inverse Pyramid shape (Fig. 3 and S5), but in most mountains globally surface area reduction due to topography hinders large benefits above 1 km. Also, at higher elevations the available surface area is scattered over more isolated mountain tops and ridges, making natural habitat connectivity a unlikely strategy for taxa or conservation efforts. So I wonder if the benefit of moving upslope and "taxa encounter more intact land area following range shifts" is indeed such a benefit considering the variety of different mountain shapes. Elaborating on that could provide an opportunity to highlight the more practical elements of their mountain by mountain analysis, relative to that which has been proposed before. Expanding the number of mountains as shown in figure 2, e.g. providing it in the supplementary information as was done for the (great) Elsen and Tingley paper, would be a very welcome contribution to the manuscript and I would expect it to increase its citability .

Here some specific comments per line:

L. 1 The topography and human pressure do not actually alter species responses to climate warming as such – taxa still move upslope – but they are likely hindered by the severely reduction in available surface area under current human presence.

L. 13 I am not overly convinced by the argument that 'Protecting and connecting intact mountain landscapes while restoring degraded landscapes may help ensure species can realize potential increases in intact land area following elevational range shifts.' Undoubtedly true but this statement feels fairly weak considering the presented figures and analysis done for each mountain region. For instance, should conservation efforts be focused on different elevations depending on the shape and the available intact surface area? Should conservation efforts connect natural area along the same elevational band or facilitate movement upwards (trans-elevational corridors within national parks)? Along lower elevations, intact habitat is likely to be scattered over a larger area surrounded by human land area and at high elevations, intact habitat is isolated on ridges and mountain tops: are natural parks fulfilling the expected need for taxa to undergo upslope shifts? Only few papers have looked into connectivity along the elevation gradient but could provide a very important addition to available surface area analysis. See for instance: Flantua et al. 2014 Missouri Botanical Garden; Bertuzzo et al. 2016 PNAS; Salles et al. 2019 Earth Surface Dynamics). However, none used an index of intact

surface area to calculate the "true" potential connectivity. Here again the usefulness of the manuscript presented here by the authors.

While the unsurprising suggestion related to the need of protecting and connecting natural mountain landscapes is not necessarily a problem, I would argue that the authors could do more to draw out the connections between their results – both conceptually and analytically – and what the implications for conservation management in differently shaped mountains are.

L. 56 Human Footprint index: Will these raster data also be made freely available?

L. 128. I am not so sure if Fig. 2 follows your argument here. The examples displayed in this figure show at least five mountains where the low elevation area changes run parallel for both scenarios, and then others where intact land area changes are lower. There seems to be more emphasis on the example of the European Alps as a "general" trend that the reader perceives from the figure. For this particular statement, it would do to just refer to Fig. 3.

L. 152. There is a reference to Fig. S2 to where it states that 24% of the mountains is entirely under intense human impact. I would then expect to see a map that show those 239 mountains but here highlighted in orange are the mountains that have changed their classification. Though related, that is not the same.

L. 157. It would be useful if the authors would give examples here of mountains of which the bases are several km above sea level. This is also useful to figure S7.

L. 226. The paper here cited (Steinbauer et al. 2018) does not look into endemism and only uses mountain summits which is a specifically small part of the mountains where taxa from different elevations are forced together into novel associations as climatic conditions change. It is not surprising that at mountain summits the total richness increases as lower elevation taxa move upslope and start occupying summits alongside the taxa already at their geographical limits. The question is still fairly unanswered if these high elevation taxa will endure (using a part of their potential niche space) or eventually perish (and we are still awaiting some sort of extinction debt). E.g., field data is contradicting the 'falling of the mountain-scenario' by many modelling studies (see Rumpf et al. Nature Comm. 4293)

L. 255 "...The most comprehensive dataset on mountain ranges currently available." For the information of the authors, there is also a dataset on mountain regions by Rahbeck et al Science 2019 and available through https://macroecology.ku.dk/resources/mountain_regions/. The mountain dataset shows high resemblance with the mountain inventory first described in the bookchapter by Fjeldsa (same author) in Hoorn et al. (2018; Mountains, Biodiversity, and Climate). Their polygons are coarser though than the dataset used by the authors and high elevation plateaus do not seem to classify as a mountain area in their dataset (that's why the Central Andes has a hole in the middle of the mountain). No need to add as a citation but just for info.

L. 322+327. Within the same paragraph the authors are using "scenario" for different elements of your analysis. Scenarios of with and without land area with humans, and the scenario of climate change. Ideally find a different wording for the first.

L. 346. There is no Fig. 3C. I guess the authors refer to the background map of fig. 2? Then why would this be a heatmap of "expected" changes?

L. 376. It would be useful if the black polygons would have the names or numbers to link to the mountain figures to the right.

L. 386. The lower bar of this figure states as title "Proportion of elevational gradient where the change in Area[intact] is higher or equal to the change in Area[total]". But the color bar is actually indicating the proportion of change comparing intact and total area at the scale of the total mountain range, correct? This is somewhat confusing.

L. 389. In the inset graph, what does it mean when the change in area over intact land levels off completely at -100%? Did species then run completely out of available intact land? The pattern is quite pronounced in the Anti-Atlas range and Sierra Nevada in Mexico, but the implications are not

clear.

L. 389. I am wondering about this pattern in the European Alps and Karakorum mountains where the proportion along lower elevations show increases, but many examples show overall decreases. The much lower available surface area in km² (Fig. 1) causes fluctuations in percentages to be quite abrupt (e.g. Karakorum where only a few km² is available up to 3000 masl). The proportion of intact land area remaining in mountains around the world seems to receive less attention than the percentage changes (comes also later in the manuscript as fig S6) but it actually has important implications for the percentage values in figure 2 for instance. Surface areas changes in percentages can easily show large changes when dealing with very low intact surface areas, e.g. the lower elevations of the European Alps with less than 20 km along certain elevations? (Fig. 1c). >200 % changes towards higher elevations then seems to picture the situation brighter than it truly is. This aspect has received little attention by the authors, though my intention is not to invalidate the usefulness of the main figures of the paper.

L. 416 I would suggest the authors to colour differently the country boundaries in b or make them less thick or remove them completely.

L. 537. A part of the reference seems mistaken? Steinbauer.....Nature, A. K. & Medeiros, R.? The reference is also missing the year.

Codes in supplementary info on codes: Example dataset works perfect!

Much looking forward to the next version of the manuscript.

Kind regards,

Suzette Flantua

Response to Reviewers

Reviewer #1 (Remarks to the Author):

Manuscript review: Topography and human pressure in mountain ranges alter expected species responses to climate change

Recommendation: Minor revision

COMMENT 1: Summary: This paper covers an important aspect of species responses to global warming in montane regions by analyzing changes in area with elevation in mountain ranges of different geometric configurations. The overall topic should be interest to ecologists, conservation biologists, and some earth scientists.

The analysis combines changes in area along with the Human Footprint metric of human impacts on land use and arrives at two important conclusions. First is that the hypsometric shape of mountain ranges differs from the shape when constrained by the area of “intact” land area by a measure of human pressure. Second is that, in some mountain ranges, species moving upslope will encounter more “intact” land area than at lower elevation, thereby gaining potential habitat. These are worthwhile insights. However, the authors could strengthen the paper by recognizing some of the complexities that are glossed over or receive no mention at all. Also, some of the phrasing could be usefully clarified.

RESPONSE 1: Thank you for your helpful comments and suggestions for ways to strengthen and improve the overall clarity of our manuscript. We have revised the manuscript and expanded the discussion following the suggestions below. Please see below for details.

COMMENT 2: Explain in more detail what “human pressures” involve; which of these have the greatest impact on montane biodiversity?

RESPONSE 2: We now explicitly state that human pressures involve activities of resource extraction, infrastructure development, and habitat conversion in L31-35 of the Introduction, highlight that the subset of habitat conversion activities (agriculture, pasture, and croplands) is extensive in mountainous regions, and provide added citations:

L31-35: “Human pressure on montane landscapes is predominantly through resource extraction, infrastructure development, and habitat conversion. Indeed, habitat conversion for agriculture, pasture, and cropland is extensive in mountainous regions¹², is a leading driver of biodiversity loss globally¹³, and is often expected to exacerbate the negative effects of climate change¹⁴.”

COMMENT 3: The phrase “intact land area” is misleading. “Intact” has multiple meanings, one being that a specific parcel of land retains most or all of its native ecosystem diversity and functions; another is that land is spatially connected (e.g., the opposite of fragmented). However, even under stable climates, similar habitats in montane regions are fragmented and sometimes

widely separated by the topography itself. This second meaning of intact does not apply well to montane regions. Some other term would be useful to avoid ambiguity.

RESPONSE 3: We explored other adjectives to contrast with areas under intense human pressure, but found the alternative word choices equally problematic. To reduce ambiguity in our intended meaning of the word, we now define our use of intact in L55-58 in the Introduction:

L55-58: “Here, we adopt the definition of mountain ranges used by the Global Mountain Biodiversity Assessment²⁶ and use the term ‘intact’ for areas that are not under intense human pressure that can negatively impact species persistence (see also Results and Methods).”

We reiterate this again in L61-65, stating:

L61-65: “We then reclassified mountain ranges based only on intact area, i.e. areas not under intense human pressure, by applying a threshold²⁴ to the Human Footprint index (HFI), a spatially-explicit map of weighted cumulative threat that combines eight separate direct threats from anthropogenic activities: croplands, pastures, roads, railways, navigable waterways, human population density, nighttime lights, and the built environment, circa 2009²⁶ (Fig. S1; Methods).”

Finally, in the Methods section, we now contrast the term ‘intact’ with ‘areas under intense human pressure’ in L344-352:

L344-352: “We then repeated this procedure after removing all elevation values that were derived from pixels under intense human pressure based on the HFI to assign each mountain a classification based on its distribution of intact land area over elevation. To do this, we followed previous analyses using the HFI to set a threshold that separates intact land from land under intense human pressure^{24,60,61}. Consequently, we used a threshold value of ≥ 4 —roughly equivalent to land being converted to (at least) pasture and reasonably approximating when human activities have converted land to a non-natural state—to indicate land area under intense human pressure, such that HFI values < 4 were considered intact...”

COMMENT 4: The authors do not mention habitat connectivity until the discussion; how much “intact” land area is accessible at any elevation is not part of the analysis. The authors would do well to acknowledge this topic as a limitation of their study.

RESPONSE 4: We agree that while an analysis of habitat connectivity is not the focus of our analysis, it needs greater attention in the text as an important caveat of our findings. We added a paragraph to the Discussion in L228-237 devoted to this topic and acknowledge that not explicitly considering connectivity is a limitation of our study.

COMMENT 5: Explain how “mountain range” is defined in this study, even if borrowed from Korner’s previous analysis. This is important because 1010 mountain ranges seems like a lot;

other recent studies of global mountain ranges on the continents list 200-300 ranges by different criteria.

RESPONSE 5: To clarify, we now state in L55-56 that we are adopting the definition of mountain range from the Global Mountain Biodiversity Assessment (while providing a reference) and extend the definition of mountain ranges in the Methods to better describe the approach taken by the authors to determine the boundaries of the 1,010 mountain ranges.

Comments on specific passages:

COMMENT 6: L 40: elaborate on the term “detailed topography.” What aspects of topography are included, which are excluded?

RESPONSE 6: We changed the wording of this sentence to better reflect that ‘detailed’ was meant to refer to ‘high-resolution’, in contrast to coarse resolution DEMs that may lead to inaccurate area estimates.

COMMENT 7: L 45: Explain what is meant by “human pressure.” Which activities have the greatest negative impact on native biodiversity? This topic needs elaboration within the main text.

RESPONSE 7: We now explain what is meant by ‘human pressure’ in L31-35 of the Introduction of the revised manuscript:

L31-35: “Human pressure on montane landscapes is predominantly through resource extraction, infrastructure development, and habitat conversion. Indeed, habitat conversion for agriculture, pasture, and cropland is extensive in mountainous regions¹², is a leading driver of biodiversity loss globally¹³, and is often expected to exacerbate the negative effects of climate change¹⁴.”

We also describe the individual pressures in more detail in L320-334 of the Methods, which should provide the appropriate context.

COMMENT 8: L 52: Explain the meaning of “intact” here (if authors keep this term). Must the land area under consideration be continuous (connected) in order to be habitable range? What proportion of montane species survive successfully as metapopulations?

RESPONSE 8: We now define ‘intact’ to give the intended meaning as land not under intense human pressure. See also Response 3 above.

COMMENT 9: L 56: Explain what “mountain range” means in this study.

RESPONSE 9: We now state that we follow the definition used the Global Mountain Biodiversity Assessment, provide the appropriate citation, and refer the readers to the detailed definition in the Methods section for brevity in text.

COMMENT 10: L 58: Mention here what model of climate change is used as the basis for modeling upslope range shifts. Somewhere it is worth noting that temperature is not the only important aspect of global climate change; changes in precipitation and orographic gradients are quite significant in montane regions. In some studies (e.g., Moritz et al. 2008, Science v 322), mammal species shift in different directions (up, down, not at all).

RESPONSE 10: We changed our modeling framework to use mountain range-specific temperature lapse rates and mountain range-specific rates of warming based on 17 general circulation models from two representative concentration pathways (RCP 4.5 and 8.5). We describe that these models were used to model range shifts in the ‘Constraints to available intact land area for species undergoing elevational range shifts’ section of the Results and provide expanded details in the ‘Modeled range shifts’ section of the Methods.

We also note in L176-179 that our modeling framework is meant to highlight area changes from temperature-induced range shifts, which are predominantly upslope (though not always), and acknowledge that other climatic factors influence species ranges to lead to heterogeneous range shifts. We now cite Moritz et al. 2008 and Tingley et al. 2012 here in support.

COMMENT 11: L 87 and elsewhere: “predominately” should be “predominantly.”

RESPONSE 11: We made this correction throughout.

COMMENT 12: L 90-93: explain here the different kinds of human pressure and activities and the gradient of pressure represented by your scoring system.

RESPONSE 12: We now include a definition of the Human Footprint index in the Introduction in L61-65 where we list the eight threats that are included in the composite map:

L61-65: “We then reclassified mountain ranges based only on intact area, i.e. areas not under intense human pressure, by applying a threshold²⁴ to the Human Footprint index (HFI), a spatially-explicit map of weighted cumulative threat that combines eight separate direct threats from anthropogenic activities: croplands, pastures, roads, railways, navigable waterways, human population density, nighttime lights, and the built environment, circa 2009²⁶ (Fig. S1; Methods).”

We also added expanded details on the original overall HFI scoring methods and methods for scoring for each of the threat components in the Methods in L320-334:

L320-334: “Human pressure. We obtained a previously published global map of the Human Footprint index (HFI) representing a metric of human pressure²³. The HFI is a weighted cumulative threat map at a spatial resolution of 30 arc-seconds (1 km²) that combines eight separate direct threats from anthropogenic activities, circa 2009: croplands, pastures, roads, railways, navigable waterways, human population density, nighttime

lights, and the built environment. Global maps of each threat were scaled by the original authors based on their degree of influence on the terrestrial environment: the built environment was given a value of 0 or 10 (all areas mapped as built given score of 10); human population density and nighttime lights were scaled to a continuous range of 0-10; croplands were given a value of 0 or 7 (all areas mapped as crops given score of 7); pasture was given a value of 0 or 4 (all areas mapped as pasture given score of 4); roads were scaled from 0-8; railways were given a value of 0 or 8 (all areas within 500 m of a railway given a score of 8); and navigable waterways were scaled to a range of 0-4²⁶. Scores for individual threats were then summed and weighted by the original authors to make a composite map with a range of 0-50. HFI values were then extensively validated against satellite imagery²⁶ yielding high confidence they represent conditions of human pressure.”

COMMENT 13: L 100-104: Run-on sentence.

RESPONSE 13: We separated this sentence into two sentences.

COMMENT 14: L 115-117: this passage needs rephrasing. Since the study does not include a consideration of relevant habitats (or their connectivity) for any species, then it cannot assess vulnerability to habitat loss. Discuss in terms of land area available for potential habitat.

RESPONSE 14: We changed the wording following the suggestion. The sentence now reads “this provides an indication of how more or less vulnerable a species may be to reductions in potential area in each of the mountain ranges in the future under climate change”.

More generally, we revised references to ‘habitat’ in the manuscript throughout when our actual intended meaning was intact or potential area to better align with what our study aims to assess.

COMMENT 15: L 127-128: This result needs clarification. The amount of “intact” land area must be a subset of the total land area. Thus, how can the second scenario result in a greater amount of projected “habitable” area? Explain more clearly that your results are about proportional change rather than absolute area.

RESPONSE 15: We added a sentence to L133-134 that clarifies that species more often experienced greater percentage of changes in area following range shifts in the intact land area scenario.

COMMENT 16: L 188-191: The logic of this passage is unclear. There is already a lot of agricultural activity at high elevations in tropical regions. Why would this not continue or even expand under global warming, especially as low-elevation regions become too warm or dry for successful farming?

RESPONSE 16: We expanded this sentence to explain that some future agricultural productivity models indicate that productivity is expected to decline because of changes in the timing and duration of the growing cycle and significantly reduced potential for multiple cropping:

L201-205: “Moreover, future land-use scenarios do not universally predict increasing human pressure towards higher elevations. For instance, overall agricultural productivity is projected to decline in the tropics due to changes in the timing and length of the growing cycle and significantly reduced potential for multiple cropping^{35,36}, which may reduce human pressure in tropical mountain ranges over time at all elevations.”

COMMENT 17: L 204: Habitat connectivity is mentioned here in the discussion. It seems like a crucial aspect of land area for successful range shifts.

RESPONSE 17: We agree, and we now highlight that analyzing the configuration of intact land area was not the focus of our study, is a limitation of our study, and should be addressed in future work in L228-237 of the revised manuscript:

L228-237: “We note that our focus on the amount of intact land area—and not its configuration along elevational gradients that could facilitate or hinder connectivity between intact landscapes or the physical environment³⁹⁻⁴¹—is a limitation of our study that should be addressed in future work to more accurately understand montane biodiversity responses to global change. In fact, while protected areas are disproportionately biased towards high elevations, comprise more intact landscapes²⁴, and reduce the rate of intact forest loss from timber harvest⁴², they are not evenly distributed or well-connected along elevational gradients in many cases¹⁶. Consequently, restoring degraded landscapes and strategically planning future protected areas to bolster elevational connectivity will be crucial to increase the chances that species can realize any benefits from increased intact land area at higher elevations.”

COMMENT 18: L 229: Why would ecological communities in montane regions be more tolerant of land-use change than those at low elevations? Also, correct the phrasing in this sentence: “ecological communities” are equated to “species” but these are not the same thing.

RESPONSE 18: We now explain in L254-258 that montane species evolved in more variable climates than lowland species and are thus thought to be more adapted to cope with temperature changes that arise from habitat modification (for instance, logging significantly raises the temperature in forests). We also changed the wording from “ecological communities” to “montane species”:

L254-258: “. Some evidence suggests that montane species may be more tolerant of land-use change than lowland species because they evolved in more variable climates and may thus be more adapted to cope with temperature changes arising from habitat modification (e.g., increased temperatures following logging)⁴⁶⁻⁴⁸. This could provide further rescue effects from pressure related to human activities.”

COMMENT 19: L 246-248: The last sentence of main text has unclear logic. On the one hand human population and activities are likely to increase at higher elevations; on the other hand, there is greater protection for montane ecosystems than in lowlands. It’s not clear though which

of these trends will have the greater impact on native biodiversity.

RESPONSE 19: We agree that there isn't a clear contrast and revised the text now in L293-297 to more succinctly state the broader point that integrating topography and land-use into assessments is important to accurately estimate species vulnerability to climate change:

L293-297: “More broadly, analyzing global topography and land use in concert reveals that species driven upslope in response to future warming may have access to more intact land away from intense human pressures that many montane species currently face. Conservation practitioners must integrate topography and human pressure to accurately assess species vulnerability to climate change.”

Methods section

COMMENT 20: L 251-255: Explain here are mountain ranges are recognized.

RESPONSE 20: We added to the description of the definition of mountain ranges in L302-308 of the revised manuscript to describe the process by which they are delineated by the original authors of the dataset.

L302-308: “Delineations in the dataset were made by expert evaluation using maps, atlases, and inventories of mountain ranges, and boundaries were subsequently informed by terrain ruggedness, which is a measure of the elevational change between focal and neighboring cells of a digital elevation model. Boundary delineations were optimized to maximize the inclusion of rugged terrain while minimizing nonrugged terrain. To our knowledge, it is the most comprehensive dataset on mountain ranges with the greatest number of mountain ranges delineated currently available.”

COMMENT 21: L 272-273: What does the language in parentheses mean? Explain for the reader who has not read the original Human Footprint report.

RESPONSE 21: We updated and expanded this section of the Methods in L320-334 to more explicitly describe the scoring for individual threat components and also now include the citation for the data paper associated with the human footprint dataset. See Response 12 above for associated text changes.

COMMENT 22: L 281: What is a dip value?

RESPONSE 22: We reworded this sentence to indicate that a dip value is the statistic resulting from performing Hartigan's dip test, which measures the degree of multimodality of an empirical distribution and provide a reference to the statistical test.

COMMENT 23: L 294: The “intensified” category needs explanation in the figure caption for Figure 1.

RESPONSE 23: We added the explanation of ‘intensified’ to the caption: “Mountain ranges classified as intensified have no intact land area remaining when removing all land under intense human pressure.”

COMMENT 24: L 306, 308: proportion OF land area

RESPONSE 24: We made the corrections.

COMMENT 25: L 328-329: the assumption of a fixed adiabatic temperature lapse rate is questionable, since the actual lapse rate varies around the world, especially around mountain ranges. The moist adiabatic lapse rate varies in relation with temperature (so with latitude and elevation).

RESPONSE 25: We agree and now have improved our modeling framework to use mountain range-specific adiabatic lapse rates. To calculate adiabatic lapse rates for each mountain range, we extracted all mean annual temperature data for each range from WorldClim v2.0, which matches the spatial resolution of our elevation and Human Footprint index rasters. We then fit a linear model of temperature ~ elevation within each range and used the slope of this relationship (the coefficient) as the lapse rate. We produced plots of this relationship for all 1,010 mountain ranges and now include it as Appendix B in the supplementary material.

Lapse rates ranged from -8.8 °C/km to 0.27 °C/km (mean -4.9 °C/km), with the bulk of the distribution in the range of values (-9 °C/km to -3 °C/km) reported by Minder et al. 2010 for mid-latitudes (see Comment 38 below). The figure below shows a histogram of the data for all 1,010 mountain ranges, and we also added a new supplementary figure (Fig. S5a) to the revised manuscript showing the spatial distribution of lapse rates used across mountain ranges. Indeed, this figure shows the significant heterogeneity in lapse rates you alluded to and we are pleased to include this information.

We added a new subsection to ‘Modeled range shifts’ section of the Methods section describing this approach in our modeled range shifts procedure.

Histogram of lapse rates across 1,010 mountain ranges now used in the modeled range shift exercise.

COMMENT 26: Figure 1: The area v elevation map for the Greater Caucasus looks like pyramid mountain range, even if it meets the skewness criterion for the diamond shape. Can you choose a better example? Make sure to explain what “intensified” means in the figure caption.

RESPONSE 26: We replaced the Greater Caucasus with the Bie Plateau to illustrate a better example. We also added the explanation of ‘intensified’ to the caption.

COMMENT 27: Figure S1: It’s unclear how the maps in (a) and (b) relate to each other. If the Yellow for intensified is from the same dataset in both maps, then why do so few areas show up as yellow in (a)?

RESPONSE 27: Figure S1a illustrates the original Human Footprint index that is a continuous variable with a range from 0-50. Figure S1b is a binary map that takes the values from S1a and assigns those values < 4 to dark grey (illustrating intact) and ≥ 4 to yellow (illustrating under intense human pressure). We changed the wording of S1a to indicate that it depicts the HFI as a continuous variable from 1-50 and S1b represents areas that fall above and below the threshold we used to identify areas under intense human pressure.

Catherine Badgley, University of Michigan

Reviewer #2 (Remarks to the Author):

General comments to the Editorial Board and to the authors

COMMENT 28: By focusing on mountain ecosystems worldwide, the authors aim at assessing the impact of species redistribution induced by global warming while accounting for the effects of both human pressures on the environment and the shape of each mountain range on the amount of land area that will remain available for mountainous species along the elevational gradient. This is a very important research question which has important implications for the future of mountain biodiversity. Yet, I do have serious concerns regarding the analytical framework proposed by the authors to answer this timely question. I do not have strong concerns regarding the first step of the authors' analytical framework which focuses on assigning each mountain range to a given "shape" based on either total land area or "intact" land area. My main concern is about the second step of the authors' analytical framework, when the authors rely on a single hypothetical species to assess the potential change in available "intact" area, following climate warming and isotherms' shifts along the elevational gradient, throughout all mountain ranges worldwide. This is a very strong and problematic, in my opinion, simplification of the reality. First of all, I would recommend the authors to use a multi-species approach to assess the fate of mountain biodiversity in terms of loss or gain in the available land area under future climate change. Second, I would recommend the authors to rely on species response curves along the temperature gradient rather than relying on a single species' elevational range of 800-m range whatever the mountain range. Indeed, following the authors' approach, this means that the hypothetical species may have different temperature and elevational margins from one mountain range to the other. Thus it is difficult to interpret the main findings. I would rather recommend the authors to use a set of virtual species (see Leroy et al., 2016) to create several ($n = 1000$, for instance) species response curves along the temperature gradient and then assess the overall (mean and standard deviation) change in available land area following a given climate change scenario (see WorldClim or CHELSA data for future climate change scenarios) and the subsequent shift of each of the virtual species. Using such an approach should provide much more realistic and general findings than the current findings based on a single hypothetical species: the sample size is equal to one here!

Besides, I would also warn the authors about the implications of assuming a synchronous biotic response of species to climate warming based on a $+2^{\circ}\text{C}$ warming rate and a fixed temperature lapse rate of $-6.2^{\circ}\text{C}/\text{km}$. Indeed, species are unlikely to shift as fast as the isotherms are shifting, even in mountain ecosystems. There are many clear reasons to believe that living organisms will lag behind the shifting isotherms (Alexander et al. 2018). In fact, the velocity of species range shifts for terrestrial organisms, even in mountain ecosystems, is clearly lagging behind the velocity of isotherm shifts (Lenoir et al. 2019). Thus, these kind of disequilibrium dynamics should be discussed by the authors.

Finally, I do have several more minor comments that I have listed below. I hope these comments

and suggestions together with my main recommendations will help the authors improve the quality of their research.

RESPONSE 28: Thank you for the detailed comments. We agree that capturing more realism in our modeling exercise is important to reach robust conclusions, and therefore we have significantly expanded and improved our analytical framework following the recommendations, as described in more detail below. This includes:

(1) Calculating and using mountain range-specific adiabatic lapse rates in our modeling procedure, acknowledging that a fixed adiabatic lapse rate for all ranges is an inappropriate simplification (see Responses 25 and 38).

(2) Using a suite of 17 general circulation models from WorldClim to calculate mountain range-specific average warming scenarios for two representative concentration pathways (RCP 4.5 and 8.5), acknowledging that rates of warming vary over space (see Response 10 above).

(3) Significantly expanding the number of hypothetical species considered for each mountain range, acknowledging that species have a wide range of ecologies in terms of being adapted to a range of elevations with both narrow and wide elevational distributions (see Response 36 below).

We now also discuss the need for these improvements with appropriate citations in two new Methods subsections under the ‘Modeled range shifts’ section. We also added a new paragraph to the Discussion in L259-269 regarding further caveats related to disequilibrium dynamics and range shifts lagging behind warming along with the recommended citations from Alexander et al. 2018, Lenoir et al. 2019, among others.

Overall, the changes we have made to the modeled range shifts analysis now show considerably more nuance and strengthen the overall findings by showing how the choice of elevational range size or warming scenario affects the percentage of area gains following elevational range shifts.

Specific comments to the authors

COMMENT 29: Lines 17-18: Maybe also refer to more specific research papers (not review or meta-analyses papers) that did show with hard data that animal (Moritz et al., 2008) and plant (Lenoir et al., 2008) species are shifting their range upward in mountain ecosystems as climate warms. It is fine to cite review papers and I also like Chen et al. (2011) paper, but it is also very important to acknowledge the original research papers that are at the basis of such quantitative reviews or meta-analyses. Besides, regarding review papers on climate-related range shifts, note that there are more recent ones focusing on both terrestrial and marine systems (Lenoir & Svenning, 2015; Lenoir et al., 2019).

RESPONSE 29: We agree it is important to acknowledge the empirical studies documenting climate-induced elevational range shifts alongside review papers, and now

include the references suggested.

COMMENT 30: Lines 26-28: Indeed, this is a very important point. Please see Graae et al. (2018) for a paper discussing the implications of fine-grained topography and short-distance escapes for species survival in mountain systems as climate warms.

RESPONSE 30: Thank you for pointing out this reference. We now include it in this sentence.

COMMENT 31: Lines 50-53: Yes, indeed, there is a clear research gap here. Yet, you may refer to Guo et al. (2018) here for an attempt to account for both climate change and land use in mountain systems on our understanding of species range shifts in response to global warming.

RESPONSE 31: Good point, we now include Guo et al., 2018 in this sentence.

COMMENT 32: Line 58: Not clear here what you mean by “species” and on which species you are focusing on to model upslope range shifts. Only in the results’ section the reader realizes that you are focusing on a single ($n = 1$) hypothetical species which elevational range is 800 m wide. This is a bit hard to trust and believe on the generalization of your findings which are based on such a strong simplification of the reality. Like I mentioned in my major concerns, you should really consider a multi-species approach for more realistic results and to also account for the uncertainty around your “single” simulation.

RESPONSE 32: We changed the wording here to indicate we are using hypothetical species in the modeling exercise. We appreciate the concern of considering a single hypothetical species in our analytical framework and have addressed this in the revised manuscript. Please see Response 36 below for the detailed response and description of our updated approach.

COMMENT 33: Lines 69-88: This very first paragraph of the result section is not bringing new findings. I would rather report these findings in the supplementary information as this is more or less a repetition of what has already been published in the scientific literature (cf. Elsen & Tingley, 2015). The new findings only start from line 89.

RESPONSE 33: We condensed this first paragraph and moved it to the end of the Introduction (L55-72) in order to bring the new findings up front while still retaining the methodological aspects that we deemed important to provide a thorough understanding of our approach:

L55-72: “Here, we adopt the definition of mountain ranges used by the Global Mountain Biodiversity Assessment²⁵ and use the term ‘intact’ to refer to areas that are not under intense human pressure that can negatively impact species persistence (see also Results and Methods). We followed existing approaches to classify mountain range topography to one of four mountain topography classes (pyramid, diamond, hourglass, and inverse pyramid) based on the statistical properties of area-elevation distributions for total area⁹. We then reclassified mountain ranges based only on intact area, i.e. areas not under intense human

pressure, by applying a threshold²⁴ to the Human Footprint index (HFI), a spatially-explicit map of weighted cumulative threat that combines eight separate direct threats from anthropogenic activities: croplands, pastures, roads, railways, navigable waterways, human population density, nighttime lights, and the built environment, circa 2009²⁶ (Fig. S1; Methods). We then modeled range shifts on all mountain ranges for an extensive set of hypothetical species based on expected temperature changes from a suite of general circulation models under two warming scenarios, assuming species could occupy all available land area in one case and that they would be restricted only to intact land area in a second case. Our approach enabled us to quantify how interactions between topography and current patterns of human pressure potentially influence the amount of intact area available for species following range shifts across the full array of elevations for all the world's mountain ranges.”

We also reorganized the order of the Results so that the section on global patterns of human pressure along elevational gradients comes first, as we felt it provided important background for subsequent sections while presenting new results. The Results section on mountain classifications is now second, and now reports new results starting and continuing from the second sentence:

L96-102: “The frequency and spatial distributions of our mountain topography classifications were consistent with previous classifications of global mountain topography using alternative data sources⁹ (Fig. 1a). Roughly 50% of ranges (507 of 1,010) were reclassified when calculations were based on the availability of intact land area (Figs. 1b, S4): pyramid mountains accounted for 17% of all mountain ranges, diamond mountains accounted for 30% of ranges, hourglass mountains accounted for 28% of ranges; and inverse pyramid mountains accounted for 2% of ranges...”

COMMENT 34: Line 89: Not clear what “intact” land area means here. I also assume that “intact” land area for a given mountain range may differ in land use to what is considered “intact” land area in another mountain range, right? If that is the case, isn't it problematic? Would it not be better to focus on a given land use type and assess how the available surface area will change for that focal land use type (e.g. forests)? By doing so, it would be easier to interpret the overall finding/pattern across all mountain ranges worldwide. Here, you are likely to mix very different land uses that are considered “intact” across the different mountain ranges.

RESPONSE 34: We now explicitly define ‘intact’ in the Introduction in L55-58 as a way to distinguish land that is not under intense pressure from human activities. We purposely chose to use the established published approaches of creating a dichotomy between ‘intact’ land and land ‘under intense human pressure’ so that we could treat all land cover types the same, rather than having to develop separate approaches for determining intact forests, intact grasslands, intact shrublands, etc., for all land cover types that occur in mountain ranges. Additional research that falls outside the scope of our study would be required to estimate metrics specific to different land cover types.

COMMENT 35: Lines 98-100: Is it a new type of mountain “shape” here?

RESPONSE 35: While not a new shape *per se*, we do treat mountain ranges that have no intact land areas as a separate classification because their area-elevation relationships are null. We clarify this in L102-104:

L102-104: “The remaining ~24% of mountain ranges had no remaining intact land area after removing all area under intense human pressure from the analysis and were classified as ‘intensified’.”

COMMENT 36: Lines 113-146: Like I wrote in my general comments, I think that using a single hypothetical species here is a very strong and wrong simplification of the reality which may potentially lead to biased results. Indeed, such a simulation does not account for the new species shifting upslope from the lowlands. Your main finding and thus the main message of the paper is clearly biased by the fact that you consider a single species which elevational range size is 800 m wide. Do you really think that we can then generalize your main finding to all mountain species based on such a strong simplification?

RESPONSE 36: We have significantly modified our modeling framework to capture more realism as suggested here. Previously, we considered a number of species equal to the number of elevational bands, each with the same 800 m elevational range (and we also separately considered the case where the species has a 1,600 m elevational range as a sensitivity analysis). Since the conclusions reached using species with only an 800 m (or 1,600 m) elevational range may not be representative of other (or most) species, we now vary the range sizes from 100-4,000 m in 100-m increments and repeat the modeling procedure as before. Using this approach, we can calculate the number of species modeled per mountain range as:

$$s_i = b_i \times (a_i \div 100) \text{ for } a_i \leq 4,000 \text{ m and} \\ s_i = b_i \times 40 \text{ for } a_i > 4,000 \text{ m}$$

where s and b are the number of species and 50-m elevational bands, respectively, and a is the amplitude (max minus minimum elevation) of mountain range i . Taking the same example as above, a mountain range extending from 1,000 to 4,000 m would have $b = 60$ elevational bands and $a = 3,000$ m amplitude for a total of $(60 \times (3,000 \div 100)) = 1,800$ species modeled. We then take the average percentage of area change per elevational band to understand the mean response (i.e., averaged across all different elevational range sizes) per mountain range, which provides a much stronger foundation for generalization. We also plot mean and standard error percentage of change in projected area following range shifts for all modeled species within elevational range size classes across all mountain ranges by mountain classification to assess the influence of elevational range size in a new figure in the revised manuscript (Fig. 4a).

This expanded procedure also enabled us to investigate how the choice of elevational range size influenced our results. Across all mountain classes, smaller elevational range sizes were associated with larger mean percentage of change in projected area gains. Gains were then generally similar for species with elevational range sizes $> 1,500$ m on diamond, hourglass, and inverse pyramid mountains, and with elevational range sizes $> 1,000$ m on pyramid

mountains. These results support the reviewer's point here in that the results do vary to some degree by the elevational range size considered. Thus, we mostly focus on the results averaged across all hypothetical modeled species (i.e., across all elevational range size classes considered), which reflect average trends.

Additionally, we investigated how the choice of elevational range size affected our calculation of the proportion of mountain ranges where the percentage of change in area of intact land equaled or exceeded the percentage of change in area of total land following modeled range shifts and present this in a new figure as well (Fig. 4b). In general, patterns were qualitatively similar across elevational range sizes on diamond, hourglass, and pyramid mountains. We found a trend for greater elevational range sizes to be associated with greater proportions for inverse pyramid mountains, and this trend was also slightly apparent for hourglass mountains. However, generally speaking, the overall mean response (i.e., response averaged over all elevational range sizes) was similar to that of a given elevational range size.

Our new approach is described in detail in the 'Modeled range shifts' section of the Methods and in a new subsection of the 'Sensitivity analyses' section of the Methods. We describe the new results related to the choice of elevational range size in L151-165 of the revised manuscript. The insets in Fig. 2 and the plots in Fig. 3a, S6, S8, S10, and Appendix A now all show means and standard errors, which are across all hypothetical species considered (rather than just one species with an 800-m elevational range). To further aid the reader in understanding our approach, we also include a schematic of our procedure in Fig. S9.

We retain the focus on montane species to avoid making additional assumptions that may not apply to lowland species. We hope you will agree that by considering a wide variety of species' responses we significantly improved the overall findings.

COMMENT 37: Line 120: Here, you wrote that each hypothetical species is distributed along a 800-m wide elevational gradient starting from the mountain range's base. This means that from one mountain range to the other, the hypothetical species as different elevational and temperature range limits, right? How does that affect your results? That is why I strongly recommend to use a multi-species approach and also a virtual species approach by building several virtual species based on thermal response curves (cf. the species niche) rather than using elevational response curves. Indeed, 1000 m a.s.l. within one mountain range in the Tropics does not mean the same for another mountain range located at higher latitudes. That is why it would be safer to rely on species response curves along the temperature gradient rather than relying on fixed species elevational range of 800 m.

RESPONSE 37: Please see response 36 above for our updated approach that now uses a large set of elevational ranges and models range shifts from each 50-m elevational band on every mountain range. By evaluating and averaging across the full suite of elevational range sizes (100 m to 4,000 m, in 100-m increments), our new approach provides a more realistic and generalizable average response that encompasses the full diversity of elevational (and thermal, by proxy) response curves.

COMMENT 38: Lines 121-122: This is another very strong assumption to use a fixed temperature lapse rate across all mountain ranges. Besides, the reference cited here (#20) is not a

very good one or relevant one to backup such a choice of focusing on a single value for the adiabatic lapse rate. There are several studies suggesting that the temperature lapse rate may vary a lot from one mountain range to the other (see references cited in Minder & Lundquist, 2010). These two authors actually wrote in their paper: “A number of observational studies using networks of temperature sensors have revealed that spatially uniform and temporally constant lapse rates of 6–6.5°C km⁻¹ are not representative of actual surface conditions over the Appalachian mountains [Bolstad et al., 1998], the European Alps [Rolland, 2003], the Qinling Mountains of China [Tang and Fang, 2006], the central Rocky Mountains [Blandford et al., 2008], and Arctic glaciers [Gardner et al., 2009]. In all these regions mean surface lapse rates differ appreciably from the often-used 6–6.5°C km⁻¹ values”. Hence, it is clear that using a single adiabatic lapse rate for all mountain ranges worldwide is another very strong simplification of the reality, which makes a lot of simplifications... That is another reason why I would strongly recommend you to use species-specific response curves along the temperature gradient rather than relying on a single fixed species elevational range and adiabatic lapse rate to estimate the future shift of mountainous species along the elevational gradient. Using temperature data under current and future conditions will allow to circumvent this issue of having to decide on a given fixed value for the adiabatic lapse rate. Besides, I would also be careful about the implications of assuming a synchronous biotic response of species to climate warming. Indeed, species are unlikely to shift as fast as the isotherms are shifting, even in mountain ecosystems. There are many reasons to believe that living organisms will lag behind the shifting isotherms (Alexander et al. 2018). In fact, the velocity of species range shifts for terrestrial organisms, even in mountain ecosystems, is clearly lagging behind the velocity of isotherm shifts (Lenoir et al. 2019). This should be acknowledged or at least discussed in the discussion section of the paper.

RESPONSE 38: We agree that using a single fixed adiabatic lapse rate is a simplification. We have now calculated mountain range-specific adiabatic lapse rates to use in our modeling framework. Please see Response 25 above where we outline the methodology in calculating and applying these lapse rates using WorldClim data. Indeed, based on our analysis, we confirm the heterogeneity of lapse rates in mountain ranges globally and found that the -6.2 °C per km rate we originally used overestimates the lapse rate of most mountain ranges. We note that we also found positive lapse rates in a few ranges, and so in these cases our modeling framework accommodates this by having species shift downward in response.

We also appreciate your comment about species potentially lagging behind shifting isotherms. Research required to make robust assumptions about the average degree of lag, and how this might vary across mountain ranges, is beyond the scope of this work. We did update our modeling framework to use mountain range-specific rates of warming based on averages across 17 GCMs using WorldClim data. We calculated rates separately for two warming scenarios (RCPs 4.5 and 8.5). In both scenarios, the modeled species ‘keep pace’ with warming, but by comparing the results from these two scenarios, we can gain insight into slower versus faster paces of range shift, and what that means for montane species under global change. In addition, we now acknowledge this issue in the Discussion in L259-269 of the revised manuscript and include the recommended references.

COMMENT 39: Lines 141-143: Ok, but what about the choice of the range limits along the elevational gradient? Did you run a sensitivity analysis for that as well?

RESPONSE 39: Yes, we originally tested species with an 800 m and 1,600 m elevational range size. Now, we test species with 100 to 4,000 m (in 100-m increments) elevational range sizes. Please see also Response 36 above that describes our new approach in detail.

COMMENT 40: Like 146: You only tested for a larger elevational range size (1600 m), right? Not for a smaller elevational range size since the elevational range size reported in the paper for the hypothetical species is already 800 m!!! Sorry, but I am confused here.

RESPONSE 40: Yes, we originally only tested 800 m and 1,600 m, but please see Response 36 for our new approach which tests species with 100 to 4,000 m (in 100-m increments) elevational range sizes.

COMMENT 41: I really enjoyed reviewing your work and I sincerely hope that my comments will be helpful to you to improve the quality of your research.

Cheers,

Jonathan Lenoir

Suggested references

Alexander et al. (2018) Lags in the response of mountain plant communities to climate change. *Global Change Biology*, 24: 563-579

Graae et al. (2018) Stay or go – how topographic complexity influences alpine plant population and community responses to climate change. *Perspectives in Plant Ecology Evolution and Systematics*, 30: 41-50

Leroy, B. et al. 2016. virtualspecies, an R package to generate virtual species distributions. *Ecography*, 39: 599-607

Lenoir et al. (2008) A significant upward shift in plant species optimum elevation during the 20th century. *Science*, 320: 1768-1771

Lenoir & Svenning (2015) Climate-related range shifts – a global multidimensional synthesis and new research directions. *Ecography*, 38: 15-28

Lenoir et al. (2019) Species better track the shifting isotherms in the oceans than on lands. bioRxiv, <https://doi.org/10.1101/765776>

Minder & Lundquist (2010) Surface temperature lapse rates over complex terrain: Lessons from the Cascade Mountains. *Journal of Geophysical Research*, 115: Issue D14

Moritz et al. (2008) Impact of a Century of Climate Change on Small-Mammal Communities in Yosemite National Park, USA. *Science*, 322: 261-264

RESPONSE 41: Thank you for your detailed comments and suggestions. We have attempted to implement all of your suggestions. We also appreciate your recommended added literature, which we have now incorporated throughout the paper. We hope you find the revised modeling framework to be significantly improved and offer much more realism with which to make sound generalizations.

Reviewer #3 (Remarks to the Author):

COMMENT 42: The paper quantifies the “true” available surface area as species range shifts move upslope along the elevational gradient in mountains due to climate warming. The available surface area is naturally affected by humans (here quantified by the Human Footprint Index) and thus the authors compare the total potential available surface area (if the entire mountain would have intact land area) vs what is truly available when human presence and thus land use are included. They use a similar approach taken by Elsen and Tingley (2015) in classifying the shape of mountains around the world, but now using the recently published dataset of mountains comprising of an inventory of all mountain ranges globally (n = 1010). They first test if mountains change their ‘Elsen and Tingley’-shape by comparing the potential (total area) vs intact surface area ([total area] – [area used by humans]), and describe in more detail an overview of mountains globally that are under high human pressure.

To demonstrate the implications of this change in surface area, the authors then model how upwards shifts of taxa would be affected by the available surface area along the elevational gradient under a 2 degrees warming scenario. They do so by modelling two types of hypothetical montane vertebrate taxa, those with a relatively short elevational range (800 vertical meters) and a wide elevational range (1800 vertical meters). The main results of the authors focus on the percentage change in projected total and intact land area for 50-m elevational bands for all mountains globally. The authors also show at which relative elevation the peak in human pressure occurs. Summarizing, their results show the intensity of intact area loss along lower elevations and the importance of taking into the account the topography of mountains and the true available surface area that can potentially be used by taxa as they move upwards under warming climate conditions.

Overall, I enjoyed very much reading this manuscript and thinking about the implications presented by the figures. The concept of total vs intact area is very intuitive and it is a very important shout out to the numerous modelling studies of montane taxa that only take into account the total potential surface area, but not what area can actually be occupied when range shifts occur. It is so intuitive that many researchers would be surprised (and perhaps frustrated) that they did not think about it before, setting a very solid ground for this manuscript to attract a high number of citations.

I think the overall results of this manuscript provides a very useful reference on the consequences of human pressure in mountains and the need to think more "3D" in terms of

mountain conservation management, but here I would like to follow up on the authors' Discussion section which is much focused on making the link to conservation and the overall implications of the results.

RESPONSE 42: Thank you for your positive feedback on our manuscript.

COMMENT 43: The emphasis of the paper and figures concentrates on comparing surface area in [percentage] values and I feel that that obscures a bit the emergency state of the remaining intact surface area along low and mid-elevation areas, and also the seemingly high number of mountains where surface area change is increasingly negative moving upslope (Fig. 3a). Most benefit of moving upslope seem to occur up to 1000 masl (Fig. 2) and for mountains with Inverse Pyramid shape (Fig. 3 and S5), but in most mountains globally surface area reduction due to topography hinders large benefits above 1 km. Also, at higher elevations the available surface area is scattered over more isolated mountain tops and ridges, making natural habitat connectivity a unlikely strategy for taxa or conservation efforts. So I wonder if the benefit of moving upslope and "taxa encounter more intact land area following range shifts" is indeed such a benefit considering the variety of different mountain shapes.

Elaborating on that could provide an opportunity to highlight the more practical elements of their mountain by mountain analysis, relative to that which has been proposed before. Expanding the number of mountains as shown in figure 2, e.g. providing it in the supplementary information as was done for the (great) Elsen and Tingley paper, would be a very welcome contribution to the manuscript and I would expect it to increase its citability.

RESPONSE 43: We appreciate this thoughtful comment on discussing more of the complexities of our work, providing caveats to our study, and offering general ways to improve our paper overall. We have now expanded the Discussion in L228-237 to discuss the limitations in terms of connectivity, acknowledging that our analysis is concerned with the amount of intact land area available for species as they undergo range shifts, but not the configuration. We also do more to highlight the large variability in mountain ranges, for example in L144-150 and L282-292 of the Discussion, to drive home the point that responses and potential benefits will be very context specific. Furthermore, following your suggestion, we now reference a new exhaustive appendix (Appendix A) in L150, which shows the results like the insets of Fig. 2 for each of the 1,010 mountain ranges so readers can appreciate that complexity, while also adding a few additional insets to Fig. 2 to balance the geographic coverage and response patterns illustrated in the main paper. We hope these changes will better illustrate that in many cases there is still significant concern strictly from the perspective of range shifts, and factoring in land use can in some cases make expectations even more dire.

Here some specific comments per line:

COMMENT 44: L. 1 The topography and human pressure do not actually alter species responses to climate warming as such – taxa still move upslope – but they are likely hindered by the severely reduction in available surface area under current human presence.

RESPONSE 44: Indeed, the expected species response is that there will be a reduction in available area that is compounded with human pressure as species move upslope. But what we find in the majority of cases is that, when you factor in how much intact land area there is currently, under climate change species may gain not only available surface area but available intact area, and even more counterintuitively may gain available intact area even if available surface area is reduced.

COMMENT 45: L. 13 I am not overly convinced by the argument that ‘Protecting and connecting intact mountain landscapes while restoring degraded landscapes may help ensure species can realize potential increases in intact land area following elevational range shifts.’ Undoubtedly true but this statement feels fairly weak considering the presented figures and analysis done for each mountain region. For instance, should conservation efforts be focused on different elevations depending on the shape and the available intact surface area? Should conservation efforts connect natural area along the same elevational band or facilitate movement upwards (trans-elevational corridors within national parks)? Along lower elevations, intact habitat is likely to be scattered over a larger area surrounded by human land area and at high elevations, intact habitat is isolated on ridges and mountain tops: are natural parks fulfilling the expected need for taxa to undergo upslope shifts? Only few papers have looked into connectivity along the elevation gradient but could provide a very important addition to available surface area analysis. See for instance: Flantua et al. 2014 Missouri Botanical Garden; Bertuzzo et al. 2016 PNAS; Salles et al. 2019 Earth Surface Dynamics). However, none used an index of intact surface area to calculate the "true" potential connectivity. Here again the usefulness of the manuscript presented here by the authors.

RESPONSE 45: Our aim is to highlight how a lack of accounting for both topography and human pressure might give incorrect assessments of species vulnerability under climate change. The topics you raise are very relevant to the discussion and we now expand our Discussion in L228-237 to discuss these issues regarding protection and connectivity, including the recommended citations and better clarify the focus of our paper. We also changed the wording of the last sentence in the abstract to better drive home our major take-home message of the need to integrate topography and human pressure to produce accurate species vulnerability assessments under climate change that often inform priorities for protection, connectivity, and restoration activities.

COMMENT 46: While the unsurprising suggestion related to the need of protecting and connecting natural mountain landscapes is not necessarily a problem, I would argue that the authors could do more to draw out the connections between their results – both conceptually and analytically – and what the implications for conservation management in differently shaped mountains are.

RESPONSE 46: We agree and have updated the last sentence to state a more powerful take-home message (see Response 45). We now also expand on this point in more detail in the Discussion, adding a new paragraph in L282-292 to draw these connections and discuss the conservation implications of our results, providing a dedicated example of how conservation actions might change if factoring in information on patterns of both topography and human pressure.

COMMENT 47: L. 56 Human Footprint index: Will these raster data also be made freely available?

RESPONSE 47: The data are freely available from the original authors as described in the data paper: Venter, O., Sanderson, E., Magrath, A. *et al.* Global terrestrial Human Footprint maps for 1993 and 2009. *Sci Data* 3, 160067 (2016) doi:10.1038/sdata.2016.67

The link to the dataset is:

<https://datadryad.org/stash/dataset/doi:10.5061/dryad.052q5>

This link was also provided in the data availability statement accompanying the paper as part of the cover letter, which we believe was not made available to reviewers. This statement would be part of the published manuscript.

COMMENT 48: L. 128. I am not so sure if Fig. 2 follows your argument here. The examples displayed in this figure show at least five mountains where the low elevation area changes run parallel for both scenarios, and then others where intact land area changes are lower. There seems to be more emphasis on the example of the European Alps as a “general” trend that the reader perceives from the figure. For this particular statement, it would do to just refer to Fig. 3.

RESPONSE 48: This is a good point. We purposely chose examples for the insets of Fig. 2 that show a variety of responses, to illustrate the complexity and diversity of interactions between topography and human pressure for species undergoing range shifts. So you are correct that referring to just Fig. 3 here would better match the text and be less confusing here; we made the change.

COMMENT 49: L. 152. There is a reference to Fig. S2 to where it states that 24% of the mountains is entirely under intense human impact. I would then expect to see a map that show those 239 mountains but here highlighted in orange are the mountains that have changed their classification. Though related, that is not the same.

RESPONSE 49: This was a mistake. We should have referred to the map in Fig. 1b, which shows the 239 mountain ranges entirely under intense human pressure in yellow. We made the correction.

COMMENT 50: L. 157. It would be useful if the authors would give examples here of mountains of which the bases are several km above sea level. This is also useful to figure S7.

RESPONSE 50: We added three examples of mountain ranges that have their bases > 2,000 m above sea level (the Altiplano, the Medicine Bow Mountains, and the Tibetan Plateau) in L83-186:

L83-86: “Furthermore, pressure is predominantly focused at the bases of mountains, which can sometimes occur thousands of meters above sea level. For example, the Altiplano in

Peru, the Medicine Bow Mountains in the United States, and the Tibetan Plateau all have their bases > 2000 m above sea level.”

COMMENT 51: L. 226. The paper here cited (Steinbauer et al. 2018) does not look into endemism and only uses mountain summits which is a specifically small part of the mountains where taxa from different elevations are forced together into novel associations as climatic conditions change. It is not surprising that at mountain summits the total richness increases as lower elevation taxa move upslope and start occupying summits alongside the taxa already at their geographical limits. The question is still fairly unanswered if these high elevation taxa will endure (using a part of their potential niche space) or eventually perish (and we are still awaiting some sort of extinction debt). E.g., field data is contradicting the ‘falling of the mountain-scenario’ by many modelling studies (see Rumpf et al. Nature Comm. 4293)

RESPONSE 51: Thank you for pointing out this mistake. We changed the text to read “plant richness” instead of endemism. We also added a new paragraph to the Discussion in L259-269 that describes the important issues you raise regarding species’ lags, disequilibrium dynamics, and extinction debts, and include the citations mentioned. This is an important point we failed to address in the original manuscript that has implications for our results.

L259-269: “Related to this issue is whether species can successfully shift their ranges fast enough to keep up with the pace of warming, and whether species that do colonize high elevation habitats can persist there. Several studies of plant and animal communities have shown that species range shifts may lag behind shifting isotherms and that such lags influence disequilibrium dynamics between colonization credits and extinction debts^{6,49,50}. While there is significant variation in lags across species owing to variation in species’ physiological and demographic responses, biotic interactions, and properties of the physical environment⁵⁰, some assessments have reported significant extinction debt is looming for montane species that is more acute for endemic and cold-adapted, high-elevation species^{51,52}. Expanding our models to incorporate disequilibrium dynamics, lagged responses, and extinction debt in future work would be an important step to ensuring realistic forecasts of extinction risk for range-shifting species.”

COMMENT 52: L. 255 “...The most comprehensive dataset on mountain ranges currently available.” For the information of the authors, there is also a dataset on mountain regions by Rahbeck et al Science 2019 and available through https://macroecology.ku.dk/resources/mountain_regions/. The mountain dataset shows high resemblance with the mountain inventory first described in the bookchapter by Fjeldsa (same author) in Hoorn et al. (2018; Mountains, Biodiversity, and Climate). Their polygons are coarser though than the dataset used by the authors and high elevation plateaus do not seem to classify as a mountain area in their dataset (that’s why the Central Andes has a hole in the middle of the mountain). No need to add as a citation but just for info.

RESPONSE 52: Thank you for commenting on this dataset. It does offer an interesting comparison, and is quite similar in many ways to the dataset used in our manuscript (Körner et al., 2017). One advantage of the mountain definitions we used is that they

incorporate ruggedness, which is a key aspect of mountains in our view. This attribute is ignored in the dataset by Rahbek et al. Another advantage of the mountain range dataset we used is that they are more finely resolved (over 1,000 polygons as opposed to 136 polygons). This is important in our context because our results have implications for conservation managers, many of whom likely operate on spatial scales much closer to those of the 1,000 polygons. Nevertheless, the Rahbek et al. dataset is an exciting tool for montane research with applications to investigating broad scale biodiversity patterns.

COMMENT 53: L. 322+327. Within the same paragraph the authors are using “scenario” for different elements of your analysis. Scenarios of with and without land area with humans, and the scenario of climate change. Ideally find a different wording for the first.

RESPONSE 53: We now change the wording to say “we modeled range shifts for a series of hypothetical montane species on each mountain range in two cases—one with and one without removing land area under intense human pressure”. Also, throughout the paper, we reserve the term ‘scenario’ when referring to ‘warming scenarios’ and use the term ‘case’ when referring to analyzing data with and without accounting for human pressure.

COMMENT 54: L. 346. There is no Fig. 3C. I guess the authors refer to the background map of fig. 2? Then why would this be a heatmap of “expected” changes?

RESPONSE 54: This was a typo, it should have referred to Fig. 3b. We made the correction.

COMMENT 55: L. 376. It would be useful if the black polygons would have the names or numbers to link to the mountain figures to the right.

RESPONSE 55: We added alphanumeric symbols next to the black polygons in Fig. 1a and b that correspond with alphanumeric symbols we added within the panels of Fig. 1c.

COMMENT 56: L. 386. The lower bar of this figure states as title “Proportion of elevational gradient where the change in Area[intact] is higher or equal to the change in Area[total]”. But the color bar is actually indicating the proportion of change comparing intact and total area at the scale of the total mountain range, correct? This is somewhat confusing.

RESPONSE 56: The color bar does in fact reflect the proportion of the elevational gradient of an individual mountain range where the change in area[intact] is equal to or greater than the change in area[total]. For example, it is the proportion of the elevational range where the red line is equal to or above the blue line in the inset plot. We updated the legend title to “proportion of elevational gradient per mountain range where change in Area[intact] equals or exceeds change in Area[total]” to make this distinction clearer.

COMMENT 57: L. 389. In the inset graph, what does it mean when the change in area over intact land levels off completely at -100%? Did species then run completely out of available intact land? The pattern is quite pronounced in the Anti-Atlas range and Sierra Nevada in

Mexico, but the implications are not clear.

RESPONSE 57: Yes, -100% is meant to reflect the species has no available total or intact land area remaining and faces local extinction. We have now added that clarification to the caption.

COMMENT 58: L. 389. I am wondering about this pattern in the European Alps and Karakorum mountains where the proportion along lower elevations show increases, but many examples show overall decreases. The much lower available surface area in km² (Fig. 1) causes fluctuations in percentages to be quite abrupt (e.g. Karakorum where only a few km² is available up to 3000 masl). The proportion of intact land area remaining in mountains around the world seems to receive less attention than the percentage changes (comes also later in the manuscript as fig S6) but it actually has important implications for the percentage values in figure 2 for instance. Surface areas changes in percentages can easily show large changes when dealing with very low intact surface areas, e.g. the lower elevations of the European Alps with less than 20 km along certain elevations? (Fig. 1c). >200 % changes towards higher elevations then seems to picture the situation brighter than it truly is. This aspect has received little attention by the authors, though my intention is not to invalidate the usefulness of the main figures of the paper.

RESPONSE 58: This is an important point that we agree deserves more attention in the text. We added sentences in L219-227 of the Discussion raising this issue and use the Karakorum example you point out as it perfectly highlights that percentages in some cases can mask relatively small area gains.

L219-227: “Moreover, while species undergoing range shifts may experience large percentage increases in area in many cases, it is important to note that the total amount of intact area may still be extremely small. For example, in the case of an upslope-shifting species at the base of the Karakorum Mountains, species may experience a > 200% increase in area following a range shift (Fig. 2), but still only have < 100 km² of intact area available to occupy (Fig. 1c). This is an important consideration for all mountain ranges where human activities have reduced the overall area of intact land to nearly nothing, and suggests restoration and rehabilitation activities may still be necessary for long-term sustainability.”

COMMENT 59: L. 416 I would suggest the authors to colour differently the country boundaries in b or make them less thick or remove them completely.

RESPONSE 59: We reduced the thickness of the country boundaries in Fig. S1b to match the thickness in S1a, which we think makes the general patterns in the figure more apparent.

COMMENT 60: L. 537. A part of the reference seems mistaken? Steinbauer.....Nature, A. K. & Medeiros, R.? The reference is also missing the year.

RESPONSE 60: Thank you for pointing this out. The issue was with importing the reference using citation software. We made the corrections.

COMMENT 61: Codes in supplementary info on codes: Example dataset works perfect!

Much looking forward to the next version of the manuscript.

Kind regards,

Suzette Flantua

RESPONSE 61: Thank you for checking the code! We updated the sample code to provide an example of our modeling procedure using mountain range-specific adiabatic lapse rates, mountain range-specific warming rates (for two warming scenarios), and the full set of different elevational range sizes considered.

Reviewers' Comments:

Reviewer #1:

Remarks to the Author:

In general, the revised version of this manuscript has addressed the main concerns that I raised in my review of the original draft. The overall analysis of how "intact" land area changes the opportunities for upslope movement of species ranges is an important point for global-change scientists and conservation biologists in particular to consider.

There is some confusion in the paragraph beginning on L 166. The first sentence references Figure 3 as illustrating outcomes under two RCP scenarios. However, Figure 3 does not present results for two RCP scenarios (or if it does, then it's not at all clear how it does). In the same paragraph, L 173-174, results for the RCP 4.5 scenario are mentioned in connection with Figure 2. But the Figure 2 caption states that results are presented for the RCP 8.5 scenario. Both of these passages need clarification.

There are a few grammatical errors to fix, and it would be useful to hyphenate compound adjectives (e.g., fine-scale topography) for ease of reading.

L 175: change were to was

L 295: change that to than

L 314: change spanned to extend

Catherine Badgley

Reviewer #2:

Remarks to the Author:

Dear authors,

Many thanks for a thoroughly revised version of the manuscript I initially reviewed. You did an outstanding job to address most of my main concerns as well as concerns from the other two reviewers. I am very much impressed by the quality of this revised version. I especially appreciate the fact that you now account for specific adiabatic lapse rates and warming rates for each of the ca. 1000 mountain ranges you studied here. This is a very nice addition that definitely increases the realism of your findings. I also appreciate the fact that you generated a large set of hypothetical species for each mountain range by basically testing almost all possible elevational range sizes. Yet, I think it would have been more powerful to use a niche-modelling approach (rather than the elevational-range focus you used) by directly generating several species' thermal niches varying in niche width and niche optimum position. This would avoid to adapt the set of hypothetical species to each specific mountain range. Anyway, the approach you used is also fine and perfectly sound to me.

I just have one remaining concern about the formulas you provided in the Methods section (lines 392-393). Maybe I misunderstood something in your explanations but I am afraid that there is a mistake in your formula, at least the formula line 392, when the focal mountain range has an elevational range lower than 4000 m. Indeed, if I take a very extreme but simple example of a mountain range starting at 1000 m and ending at 1200 m, then the amplitude (a) is 200 m and the number of 50-m elevational band (b) is 4. According to your formula, the total number of species with elevational ranges of 100 m and 200 m you can fit in that mountain range is 8 species. Yet, if I try to do it by making a quick drawing, I can only fit 4 species maximum: 1 species with a 200-m elevational range (1000-1200) as it is not possible to fit more species on such an elevational gradient which ranges from

1000 to 1200 and then 3 species with an elevational range of 100 m but distributed every 50 m along the 200-m elevational gradient (1000-1100; 1050-1150; 1100-1200), according to your explanations in the paragraph above that formula. Same if I try another simple example of a mountain range that goes from 1000 m to 1300 m, then $a = 300$ m and $b = 6$ 50-m elevational bands. According to your formula it would be 18 species in total but when I draw the diagram, I can only fit 9 species (1000-1300; 1000-1200; 1050-1250; 1100-1300; 1000-1100; 1050-1150; 1100-1200; 1150-1250; 1200-1300) according to your explanations, again half of the value I get with your formula. Could you please check your formula again? Maybe you simply need to divide it by 2 to get the numbers correct. Otherwise, if the formula you provided is the correct one, then I really miss an important information from your explanations to understand it and how you can fit, for instance, 18 species along an elevational gradient starting at 1000 m and ending at 1300 m. I can only fit 9 based on my understanding of your explanations but maybe I am wrong and misunderstood something important...

Last but not least, I think that you need to be very careful when referring to "modelling range shifts" (lines 124, 137, 140, etc.) throughout the main text and the methods. Indeed, you did not really model the expected range shifts but you rather modelled "the expected species elevational range shift under the assumption that species will closely track the shifting isotherms". This is an important detail to mention in the text as it is not exactly the same thing. That is actually the main reason why then, in the discussion section, you mention the lagging effects and the disequilibrium dynamics that are not accounted for here. Thus, the reader need to be reminded of that important detail throughout the manuscript such that he/she keeps in mind that you are not modelling species range shifts per se but rather how fast the isotherms are shifting within each focal mountain range and how this will affect the surface area available under your two study cases (total area vs. intact area). I suggest that you carefully check the text throughout the manuscript and replace "modelling species range shifts" by a more appropriate term like "Modelling species elevational range shifts under the assumption that species will closely track the shifting isotherms" or alike, for the sake of clarity.

Lines 142-143: This sentence sounds weird and awkward. I would reformulate.

Line 432: Maybe you should make it clearer here that the delta Area metric you computed is a percentage value (as indicated in the equation, line 426).

Line 436: And per mountain type (diamond, hourglass, pyramid, inverse), no?

Again, Thanks a lot for carefully considering my comments during the initial round of review. This is very much appreciated.

Best,

Jonathan Lenoir

Reviewer #3:

Remarks to the Author:

The authors addressed all comments well and the expanded discussion and analyses (e.g. lapse rate variation) are much appreciated. Overall the manuscript improved in clarity and completeness, with better explanations and examples. R codes are clear and well organized.

No further comments from my side besides looking forward to seeing the manuscript published.

Response to Reviewers

REVIEWERS' COMMENTS:

Reviewer #1 (Remarks to the Author):

Comment 1: In general, the revised version of this manuscript has addressed the main concerns that I raised in my review of the original draft. The overall analysis of how “intact” land area changes the opportunities for upslope movement of species ranges is an important point for global-change scientists and conservation biologists in particular to consider.

There is some confusion in the paragraph beginning on L 166. The first sentence references Figure 3 as illustrating outcomes under two RCP scenarios. However, Figure 3 does not present results for two RCP scenarios (or if it does, then it’s not at all clear how it does). In the same paragraph, L 173-174, results for the RCP 4.5 scenario are mentioned in connection with Figure 2. But the Figure 2 caption states that results are presented for the RCP 8.5 scenario. Both of these passages need clarification.

Response 1: The parenthetical in L169-170 refers the readers to Figures 3 and Supplementary Figure 6, where Fig. 3 shows the results using RCP 8.5 and Supplementary Figure 6 shows the results using RCP 4.5. We updated the caption of Fig. 3 to refer readers to Supplementary Figure 6 for the analogous plot using RCP 4.5 (the caption of Supplementary Figure 6 already referred the reader to the analogous plot using RCP 8.5).

The parenthetical in L176 refers readers to Fig. 2 and Supplementary Figure 7, where, like above, Fig. 2 shows the results using RCP 8.5 and Supplementary Figure 7 shows the results using RCP 4.5. We updated the caption of Fig. 2 to refer readers to Supplementary Figure 7a for the analogous map plot using RCP 4.5 and updated the caption of Supplementary Figure 7 to refer readers to Fig. 2 for the analogous map plot using RCP 8.5.

Comment 2: There are a few grammatical errors to fix, and it would be useful to hyphenate compound adjectives (e.g., fine-scale topography) for ease of reading.

L 175: change were to was

L 295: change that to than

L 314: change spanned to extend

Catherine Badgley

Response 2: We made the suggested changes in L175 and L314, but did not change ‘that’ in L295 as it was our intended meaning. We also hyphenated compound adjectives to ease reading as suggested.

Reviewer #2 (Remarks to the Author):

Comment 3: Dear authors,

Many thanks for a thoroughly revised version of the manuscript I initially reviewed. You did an outstanding job to address most of my main concerns as well as concerns from the other two reviewers. I am very much impressed by the quality of this revised version. I especially appreciate the fact that you now account for specific adiabatic lapse rates and warming rates for each of the ca. 1000 mountain ranges you studied here. This is a very nice addition that definitely increases the realism of your findings. I also appreciate the fact that you generated a large set of hypothetical species for each mountain range by basically testing almost all possible elevational range sizes. Yet, I think it would have been more powerful to use a niche-modelling approach (rather than the elevational-range focus you used) by directly generating several species' thermal niches varying in niche width and niche optimum position. This would avoid to adapt the set of hypothetical species to each specific mountain range. Anyway, the approach you used is also fine and perfectly sound to me.

Response 3: We are pleased to hear that the new adiabatic lapse rate and warming rate analyses are satisfactory and appreciated. We were intrigued by the suggestion of using a niche-modelling approach and feel this could make an excellent follow-up study to pursue, but are glad that you approve of the approach we employed varying elevational range sizes.

Comment 4: I just have one remaining concern about the formulas you provided in the Methods section (lines 392-393). Maybe I misunderstood something in your explanations but I am afraid that there is a mistake in your formula, at least the formula line 392, when the focal mountain range has an elevational range lower than 4000 m. Indeed, if I take a very extreme but simple example of a mountain range starting at 1000 m and ending at 1200 m, then the amplitude (a) is 200 m and the number of 50-m elevational band (b) is 4. According to your formula, the total number of species with elevational ranges of 100 m and 200 m you can fit in that mountain range is 8 species. Yet, if I try to do it by making a quick drawing, I can only fit 4 species maximum: 1 species with a 200-m elevational range (1000-1200) as it is not possible to fit more species on such an elevational gradient which ranges from 1000 to 1200 and then 3 species with an elevational range of 100 m but distributed every 50 m along the 200-m elevational gradient (1000-1100; 1050-1150; 1100-1200), according to your explanations in the paragraph above that formula. Same if I try another simple example of a mountain range that goes from 1000 m to 1300 m, then $a = 300$ m and $b = 6$ 50-m elevational bands. According to your formula it would be 18 species in total but when I draw the diagram, I can only fit 9 species (1000-1300; 1000-1200; 1050-1250; 1100-1300; 1000-1100; 1050-1150; 1100-1200; 1150-1250; 1200-1300) according to your explanations, again half of the value I get with your formula. Could you please check your formula again? Maybe you simply need to divide it by 2 to get the numbers correct. Otherwise, if the formula you provided is the correct one, then I really miss an important information from your explanations to understand it and how you can fit, for instance, 18 species along an elevational gradient starting at 1000 m and ending at 1300 m. I can only fit 9 based on my understanding of your explanations but maybe I am wrong and misunderstood something important...

Response 4: Thank you for pointing out this issue. Indeed, our approach in the last revision included species with ranges extending beyond the gradient (see full explanation below). Whereas the approach you employed is more intuitive and realistic, so we revised our approach.

Here is a schematic of what we did previously:

In this example of a mountain range with an amplitude of 500 m and 10 elevational bands, we calculated $10 \times (500 / 100) = 50$ species, with ranges varying in 100 m increments and distributed every 50 m as depicted above (different colored bars reflect different species). We had included those species whose ranges extended beyond the elevational gradient (i.e., those with upper ranges in grey). In area calculations, the effect was that the ranges were truncated, such that there is no area counted in any of the greyed portion of the elevational range. We had done it this way for simplicity in modeling. However, we realize now that this effectively ‘duplicates’ some species in the analysis (for example, the last five species in the above example would all have the same elevational ranges). This approach would represent more species packing at upper elevations, which could influence the statistics we calculated of mean and standard error responses.

We revised our approach based on your comment to remove those species that would have had their ranges extend beyond the elevational gradient. Doing it this way, the equation for the number of species is as the reviewer suggests:

$$s_i = b_i \times (a_i \div 100) \div 2 \text{ for mountain ranges with an even number of elevational bands}$$

$$s_i = (b_i - 1) \times ((a_i + 50) \div 100) \div 2 \text{ for mountain ranges with an odd number of elevational bands}$$

This translates into the following for the example above:

This corresponds to all possible species with elevational ranges at least 100 m, in 100-m increments, starting from every 50-m elevational band.

Using this approach, we recalculated all of our statistics and recreated all of our figures that relied on the modelled elevational range shifts assuming species closely track shifting isotherms. This included Figs. 2-4, Supplementary Figures 6-10, and Supplementary Data 1 (formerly Appendix A).

Importantly, the results and our conclusions have not changed qualitatively. For example, comparing the proportion of the elevational gradient where the percentage of change in area of intact land equaled or exceeded the percentage of change in area of total land following range shifts using our previous approach and the updated one amounted to a mean absolute proportional difference of 0.032 across all mountain ranges. Furthermore, visually inspecting the results in Figs. 2 and 3 show that results are very similar. The effect of including all species in our previous revision was most pronounced for hourglass mountains, where the inclusion had increased the confidence (narrowed the confidence intervals) and biased the mean calculations somewhat lower compared to now excluding those species (see Figs. 3a and 4a for example, to compare with the previous revision).

We also found a small but notable difference in the results of the analysis of how range size influences the proportion of mountain ranges where the change in intact area equals or exceeds the change in total area (Fig. 4b). Whereas previously the mean responses across all species were tightly correlated with nearly all range sizes for all mountain topography classes, we now see a bit more variation across range sizes and note that the previously observed signal of species with smaller elevational ranges having lower values for this statistic (primarily at lower elevations) is true for all mountain classes (not just hourglass and inverse pyramid mountains, as previously reported), and especially for hourglass and pyramid mountains. We have revised the text in L163-167 to reflect this:

“We found that the choice of modeled elevational range size influenced the overall proportion of mountain ranges where the percentage changes in area of intact land equaled or exceeded the percentage changes in area of total land (Fig. 4b). In general, species with smaller modeled elevational range sizes tended to decrease this proportion at lower elevations, particularly for hourglass and pyramid mountains (Fig. 4b).”

The overall mean responses are very similar to before (with the exception of the portion of the elevational range between 5,000-7,000 m on Hourglass mountains), which gives confidence in our overall results.

Comment 5: Last but not least, I think that you need to be very careful when referring to “modelling range shifts” (lines 124, 137, 140, etc.) throughout the main text and the methods. Indeed, you did not really model the expected range shifts but you rather modelled “the expected species elevational range shift under the assumption that species will closely track the shifting isotherms”. This is an important detail to mention in the text as it is not exactly the same thing. That is actually the main reason why then, in the discussion section, you mention the lagging effects and the disequilibrium dynamics that are not accounted for here. Thus, the reader need to be reminded of that important detail throughout the manuscript such that he/she keeps in mind that you are not modelling species range shifts per se but rather how fast the isotherms are shifting within each focal mountain range and how this will affect the surface area available

under your two study cases (total area vs. intact area). I suggest that you carefully check the text throughout the manuscript and replace “modelling species range shifts” by a more appropriate term like “Modelling species elevational range shifts under the assumption that species will closely track the shifting isotherms” or alike, for the sake of clarity.

Response 5: We have changed the text to acknowledge our assumption of species closely tracking shifting isotherms.

In L131-133 we now state: “For the purposes of our analysis, our modeled range shifts operate on the assumption that species will closely track shifting isotherms and therefore do not explicitly address lagged responses or disequilibrium dynamics (but see Discussion).”

In L140-142 we now state: “For example, modeled range shifts where species closely track shifting isotherms in the European Alps in the first case led to continued reductions in projected area available for species as total land area tended to decline monotonically with elevation”.

We also added a new clause (underlined here) in L265-268 stating: “Our analysis assumes species will closely track shifting isotherms, but several studies of plant and animal communities have shown that species range shifts may lag behind shifting isotherms and that such lags influence disequilibrium dynamics between colonization credits and extinction debts.”

We hope these changes will reinforce the assumptions of our analysis.

Comment 6: Lines 142-143: This sentence sounds weird and awkward. I would reformulate.

Response 6: We reworded this sentence as: “At higher elevations (> 2,500 m), species encountered area reductions in both cases owing largely to topographic constraints.”

Comment 7: Line 432: Maybe you should make it clearer here that the delta Area metric you computed is a percentage value (as indicated in the equation, line 426).

Response 7: We reworded this sentence as: “We contrasted the results of the two cases by separately plotting the average percentage of change in projected total area ($\Delta\text{Area}_{\text{total}}$) and the average percentage change in intact land area ($\Delta\text{Area}_{\text{intact}}$) across all hypothetical species over elevation, separately for each mountain range and also by mountain classification.”

Comment 8: Line 436: And per mountain type (diamond, hourglass, pyramid, inverse), no?

Response 8: Yes, we made the distinction (see previous comment and response).

Again, Thanks a lot for carefully considering my comments during the initial round of review. This is very much appreciated.

Best,

Jonathan Lenoir

Reviewer #3 (Remarks to the Author):

Comment 9: The authors addressed all comments well and the expanded discussion and analyses (e.g. lapse rate variation) are much appreciated. Overall the manuscript improved in clarity and completeness, with better explanations and examples. R codes are clear and well organized.

No further comments from my side besides looking forward to seeing the manuscript published.

Response 9: Thank you for again for the positive feedback and for checking the R code.